# Spring-water temperature suggests widespread occurrence of Alpine permafrost in pseudo-relict rock glaciers

Luca Carturan[1], Giulia Zuecco[1,2], Angela Andreotti[1], Jacopo Boaga[3], Costanza Morino[1], Mirko Pavoni[3], Roberto Seppi[4], Monica Tolotti[5], Thomas Zanoner[4], Matteo Zumiani[6]

[1]Department of Land, Environment, Agriculture and Forestry, University of Padova, Legnaro, Italy

[2]Department of Chemical Sciences, University of Padova, Padova, Italy

[3]Department of Geosciences, University of Padova, Padova, Italy

[4]Department of Earth and Environmental Sciences, Pavia, Italy

[5]Fondazione Edmund Mach - Istituto Agrario San Michele All'Adige, S. Michele all'Adige, Italy

[6]Geological Service, Autonomous Province of Trento, Trento, Italy

*Correspondence to:* Luca Carturan (luca.carturan@unipd.it)

**Abstract**

Runoff originating from ground ice contained in rock glaciers represents a significant water supply for the lowlands. Pseudo-relict rock glaciers host patchy permafrost, but appear to be visually relict, and therefore can be misinterpreted by using standard classification approaches. Permafrost content, spatial distribution and frequency of this type of rock glaciers are poorly known. Therefore, identifying pseudo-relict rock glaciers that might still host permafrost, and potentially ice, is crucial for understanding their hydrological role in a climate change context.

This work analyses rock-glacier spring-water temperature in a 795 km$^2$ catchment in the Eastern Italian Alps to understand how many rock glaciers classified as relict could have spring-water temperature comparable to active or transitional rock glaciers, as possible evidence of their pseudo-relict nature. Spring-water temperature, often used as auxiliary to other approaches for specific sites, was used for a preliminary estimate of the permafrost presence in 50 rock glaciers classified as relict. In addition, we present electrical resistivity tomography (ERT) results on two relict rock glaciers with opposing spring-water temperature and surface characteristics to constrain spring-water temperature results at local scale.

The results show that about 50% of rock glaciers classified as relict might be pseudo-relict, thus potentially containing permafrost. Both supposedly relict rock glaciers investigated by geophysics contain frozen sediments. The majority of cold springs are mainly associated with rock glaciers with blocky and sparsely vegetated surface, but geophysics suggest that permafrost may also exist in rock glaciers below 2000 m a.s.l., entirely covered by vegetation and with spring-water temperature up to 3.7°C. We estimate that pseudo-relict rock glaciers might contain a significant portion (20%) of all the ice stored in the rock glaciers in the study area. These results highlight the relevance of pseudo-relict rock glaciers in periglacial environments. Even if not a conclusive method, spring-water-temperature analyses can be used to preliminarily distinguish between relict and pseudo-relict rock glaciers in wide regions.

## 1. Introduction

Timings and magnitude of cryosphere runoff have high climatic sensitivity and are impacted by the current changes of Earth's climate (Engelhardt et al., 2014; Zemp et al., 2015; Carturan et al., 2019). Moreover, a deterioration of the water quality has been reported for springs fed by melting permafrost (Thies et al., 2013; Ilyashuk et al., 2014). Due to glacier decline, in the last decades growing attention has been given to other water reservoirs, such as subsurface ice, including debris-covered glacier ice and, in particular, ground ice stored in periglacial landforms such as rock glaciers and glacial-permafrost composite landforms (e.g., Brighenti et al., 2019; Jones et al., 2019; Schaffer et al., 2019; Seppi et al., 2019; Wagner et al., 2021). Projection of ice loss rates indicates that in the second half of the 21$^{st}$ century more subsurface ice may be preserved than glacier surface ice because of their different response times to atmospheric warming (Haeberli et al., 2017). Subsurface ice is therefore expected to significantly contribute to stream runoff under future climate warming (Janke et al., 2015, 2017).

Jones et al. (2018) assessed the importance of ice contained in rock glaciers at global scale, estimating that $62.02 \pm 12.40$ Gt of ice is contained in intact rock glaciers. With the adjective 'intact' we refer to the traditional categorization of rock glaciers, which distinguish between intact rock glaciers (containing ice) and relict rock glaciers (not containing ice). According to the most updated classification (RGIK, 2023), rock glaciers should be categorized into active, transitional and relict, referring exclusively to the efficiency of sediment conveyance (expressed by the surface movement) at the time of observation. This classification should not be used to infer any ground ice content.

Even though relict rock glaciers should not contain ice (Haeberli, 1985; Barsch, 1996), more recent studies showed that some relict rock glaciers can preserve permafrost and ice far below the regional lower limit of discontinuous permafrost (e.g., Delaloye, 2004; Strozzi et al., 2004; Lewkowicz et al., 2011; Bollati et al., 2018; Colucci et al., 2019). This evidence raises the question whether a significant fraction of rock glaciers classified as relict is actually to be considered 'pseudo-relict', i.e. *"rock glaciers which appear to be visually relict but still contain patches of permafrost"* (Kellerer-Pirklbauer et al., 2012; Kellerer-Pirklbauer, 2008, 2019). This question is relevant because landforms classified as relict in some regions can be up to an order of magnitude larger and more numerous than active and transitional rock glaciers (e.g., Seppi et al., 2012; Scotti et al., 2013; Kofler et al., 2020), with potentially significant ecological and hydrological impacts (e.g., Brenning, 2005a; Millar and Westfall, 2019; Brighenti et al., 2021; Sannino et al., 2021). According to Jones et al. (2019), identifying and establishing the activity state of rock glaciers is an important initial step in determining their potential hydrological significance.

Previous investigations on the possible permafrost content of relict rock glaciers looked at single case studies or small groups of landforms (e.g., Delaloye, 2004;  Kellerer-Pirklbauer et al., 2014; Popescu, 2018; Colucci et al., 2019; Pavoni et al., 2023). Studies considering a larger number of relict rock glaciers, at the regional scale, were mainly focussed on the past distribution of mountain permafrost and on the reconstruction of related paleoclimatic conditions (e.g., Frauenfelder et al., 2001; Seppi et al., 2010; Charton et al., 2021; Dlabáčková et al., 2023).

As a result, the actual distribution, frequency, and ice content of pseudo-relict rock glaciers might be underestimated, with the latter being essential for implementing worldwide estimates of water resources stored in periglacial landforms (e.g., Jones et al., 2018). Detailed geophysical investigation of selected landforms is certainly suitable as a first step towards a better knowledge of pseudo-relict rock glaciers and their ice content. However, due to logistic constraints, this approach cannot be applied to a large number of rock glaciers at the catchment or regional scale. A recent and commendable advance on this topic has been achieved by the proposition of operational guidelines on the InSAR-based

kinematic characterization of rock glaciers (Bertone et al., 2022), which can be used for thorough studies of wide areas. However, this approach is not suitable for distinguishing between relict and pseudo-relict rock glaciers, because their surface has no movement or the movement is very slow and in the same range as the uncertainty of the method.

A possible way to investigate the presence of permafrost in these landforms over large areas is by analysing spring-water temperature measured downslope of rock glaciers. Haeberli (1975) proposed the monitoring of spring-water temperature in late summer as useful evidence of permafrost, and various authors employed such method as auxiliary permafrost evidence (e.g., Frauenfelder et al., 1998; Scapozza, 2009; Imhof et al., 2000; Strozzi et al., 2004; Cossart et al., 2008). Carturan et al. (2016) demonstrated that this method can be used successfully for mapping permafrost distribution at the catchment scale. All these works are based on the evidence that, in late summer, spring water affected by permafrost has lower temperature compared to those unaffected, with upper thresholds ranging between 0.9 and 1.1°C for probable permafrost, and between 1.8 and 2.2°C for possible permafrost.

In this work, we analyse the spatial variability of spring-water temperature in a 795 km$^2$ catchment located in the Eastern Italian Alps, where 338 rock glaciers were inventoried (Seppi et al., 2012), to better understand permafrost distribution. We hypothesise that a significant portion of rock glaciers classified as relict have spring-water temperature comparable to those of active and transitional rock glaciers, as possible evidence of their permafrost content and of their pseudo-relict nature. The specific objectives of this study are to:

 i) analyse the influence of topographic and geomorphological factors on spring-water temperature,

ii) investigate the main controls on water temperature for springs downslope of rock glaciers, and particularly of relict rock glaciers,

 iii) investigate via geophysical analyses the presence of permafrost in two rock glaciers selected for their different spring-water temperature and surface characteristics, to constrain spring-water temperature results at local scale,

iv) preliminarily estimate and compare the ice content of rock glaciers and glaciers in the study area.

**2. Study area**

Val di Sole is located in the upper part of the Noce River catchment, a tributary of the Adige River, which is the main river system in northeastern Italy (Fig. 1). The catchment is 795 km$^2$ wide, with elevation ranging between 520 m a.s.l. at the outlet (Mostizzolo) and 3769 m a.s.l. at the summit of Mt. Cevedale, averaging 1705 m a.s.l. (Fig. 1). Metamorphic rocks (mica schists, paragneiss and orthogneiss) prevail in the northern side of the valley, whereas tonalite is found in the southwestern part and dolomites and limestones prevail in the southeastern part (Dal Piaz et al., 2007; Martin et al., 2009; Chiesa et al., 2010, Montrasio et al., 2012).

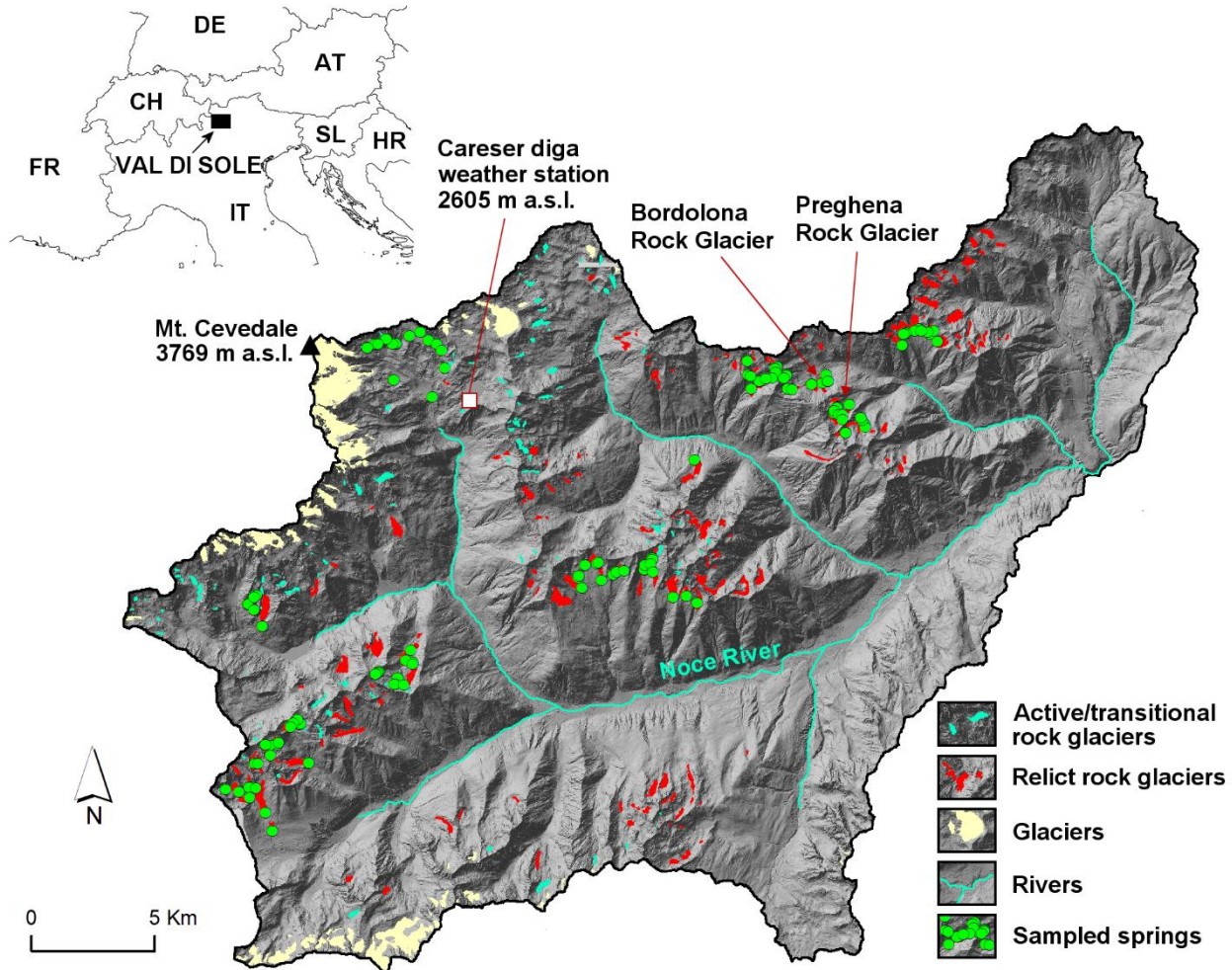

104

Figure 1: Geographic location of the study area and of sampled springs. The background is the hillshaded Lidar 2014
DEM surveyed by the Provincia Autonoma di Trento (https://siat.provincia.tn.it/stem/).

The catchment includes a glacierised area of 16 km$^2$ (in 2006, Salvatore et al., 2015). Bare bedrock and debris are found
outside the glaciers down to an elevation of 2700 m, which is the lower regional limit of discontinuous permafrost
(Boeckli et al., 2012). A discontinuous cover of alpine meadows and shrubs is present between 2200 m and 2700 m, while
below 2000-2200 m forests are dominant. The valley bottom is covered by cultivations and settlements.

Val di Sole lies in a transition zone between the "inner dry alpine zone" in the north (Frei and Schär, 1998) and the wetter
area under the influence of the Mediterranean Sea in the south. At the valley floor, the mean annual precipitation in the
period between 1971 and 2008 is ~900 mm. Precipitation increases with elevation and in the southern part, with a
maximum of 1500 mm in the Adamello-Presanella Group (Carturan et al., 2012; Isotta et al., 2014). The mean annual
0°C isotherm is located at 2500 m. The mean annual air temperature variability is dominated by elevation, whereas
latitudinal and longitudinal variations are negligible.

Seppi et al. (2012) mapped 338 rock glaciers in Val di Sole. Based on evidence visible in the orthophotos and digital
elevation models (DEMs), the largest part of rock glaciers was classified as relict (229, 68% of the total), whereas only
42 of the remaining 109 can be classified as active based on multitemporal high-resolution DEMs, and the other 67 can

be considered transitional. Most rock glaciers (302, 89% of the total) are composed of deposits of metamorphic rocks in
the orographic left side of the valley.

**3. Materials and methods**
**3.1 Experimental design**
We focussed our investigations on the northern part of Val di Sole because it has a rather homogeneous lithology
(metamorphic rocks with predominant micaschists) and mean annual precipitation of 1233 mm at 2600 m (Carturan et
al., 2016). This was done to minimise the effects of different lithologies and annual precipitation on the spatial variability
of spring-water temperature, and to highlight the role of other variables related to their catchment, upslope area or upslope
rock glaciers.
To obtain statistically meaningful and generalisable results, we designed a sampling scheme for rock-glacier spring-water
temperature considering the variability of permafrost-related characteristics in the study area, namely vegetation cover
(related to ground temperature and fine debris infill), size (length, area), elevation, slope, aspect, and lithology (Barsch,
1996; Haeberli, 1985; Lambiel and Reynard, 2001; Boekli et al., 2012).
We inspected these variables, reported for each rock glacier of Val di Sole in the database of Seppi et al. (2012), using a
correlation matrix and the principal component analysis. The aim was to evaluate their possible covariance and to optimise
the number of variables to be included in the sampling scheme. The analysis revealed high positive covariance between
length and area (both related to size). Negative covariance was found between elevation and vegetation cover, and
between slope and length/area.
Based on these outcomes and considering accessibility of springs, we built a sampling scheme around four variables: i)
rock glacier activity, ii) length, iii) mean elevation, and iv) vegetation cover. The last two variables are correlated because
active and transitional rock glaciers are at high elevation and almost free from vegetation, and the opposite is true for
relict rock glaciers. Vegetation cover is probably one of the few variables that may aid at identifying rock-glacier activity
(Ikeda and Matsuoka, 2002; Strozzi et al., 2004, Kofler et al., 2020), and it can vary greatly among rock glaciers at similar
elevation.  For this reason, we kept both elevation and vegetation, applying a modification to the vegetation-cover
classification proposed by Seppi et al. (2012). We distinguish between two classes, namely 'vegetated' and 'non
vegetated' for both active/transitional and relict rock glaciers (see Table 1 for threshold values). The vegetation cover was
visually estimated in the field and in orthophotos for each rock glacier. Our sampling scheme ensured that at least one
rock glacier was sampled for each combination of variables (Table 2). The frequency distribution of rock glacier length
and mean elevation was used to identify three terciles, employed for grouping them into short-mid-long rock glaciers and
into low-mid-high elevation rock glaciers. Frequency distributions and terciles of active/transitional and relict rock
glaciers were calculated separately (Table 2).




Table 1 - Classification of active/transitional and relict rock glaciers in two different classes of vegetation cover.

| Rock glacier category | Vegetation cover class | Meaning |
|---|---|---|
| Active/transitional | Vegetated | Vegetation cover >10% |
| | Non vegetated | Vegetation cover <10% |
| Relict | Vegetated | Vegetation cover >50% |
| | Non vegetated | Vegetation cover <50% |




Table 2: Sampling scheme used for water temperature measurements at rock glaciers springs.

| Activity state | Length | Elevation | Vegetation cover | Number of sampled rock glaciers |
|---|---|---|---|---|
| Active/transitional | Short (<142 m) | Low (<2634 m) | Non vegetated | 2 |
| | | | Vegetated | none |
| | | Mid (>2634 and <2811 m) | Non vegetated | 2 |
| | | | Vegetated | none |
| | | High (>2811 m) | Non vegetated | 1 |
| | | | Vegetated | none |
| | Mid (>142 and <251 m) | Low (<2596 m) | Non vegetated | 1 |
| | | | Vegetated | none |
| | | Mid (>2596 and <2817 m) | Non vegetated | 1 |
| | | | Vegetated | 3 |
| | | High (>2817 m) | Non vegetated | 2 |
| | | | Vegetated | none |
| | Long (>251 m) | Low (<2655 m) | Non vegetated | none |
| | | | Vegetated | 1 |
| | | Mid (>2655 and <2779 m) | Non vegetated | 1 |
| | | | Vegetated | none |
| | | High (>2779 m) | Non vegetated | 3 |
| | | | Vegetated | none |
| Relict | Short (<180 m) | Low (<2267 m) | Non vegetated | 3 |
| | | | Vegetated | 4 |
| | | Mid (>2267 and <2453 m) | Non vegetated | 1 |
| | | | Vegetated | 2 |
| | | High (>2453 m) | Non vegetated | 2 |
| | | | Vegetated | 2 |
| | | Low (<2255 m) | Non vegetated | 3 |

| | | | Vegetated | 4 |
|---|---|---|---|---|
| | Mid (>180 and <340 m) | Mid (>2255 and <2425 m) | Non vegetated | 1 |
| | | | Vegetated | 2 |
| | | High (>2425 m) | Non vegetated | 2 |
| | | | Vegetated | 3 |
| | Long (>340 m) | Low (<2222 m) | Non vegetated | 1 |
| | | | Vegetated | 4 |
| | | Mid (>2222 and <2388 m) | Non vegetated | 3 |
| | | | Vegetated | 5 |
| | | High (>2388 m) | Non vegetated | 5 |
| | | | Vegetated | 3 |
| | | | **Total:** | **67** |

**3.2 Data collection**

Water temperature was measured at 220 springs, 133 of which are located downslope of rock glaciers (multiple springs were often measured downslope of the same rock glacier), 81 are located downslope of other deposits, and 8 are located in bedrock. Springs were sampled from mid-August to mid-October, after the end of the snowmelt. Most springs have been measured once per year from 2018 to 2020, and a small group of them was also measured in 2021. In these four years, the total number of single measurements is 540.

Based on the sampling scheme (Table 2), we measured spring-water temperature at 17 active/transitional rock glaciers and 50 relict rock glaciers, which corresponds to 22% of all rock glaciers existing in the study area. All variables' combinations defined for relict rock glaciers have been sampled, whereas several combinations for active/transitional rock glaciers lack samplings. This was due to the inexistence of single combinations (e.g., there are no short and vegetated active/transitional rock glaciers at low elevation) or to the lack of springs and inaccessibility of some rock glaciers.

Measurements of spring-water temperature were carried out using a WTW Cond3310 (WTW GmbH, Weilheim, Germany) and a Testo 110 (Testo AG, Lenzkirch, Germany). These instruments have both 0.1°C resolution, but the WTW has higher accuracy ($\pm 0.1$°C) compared to the Testo ($\pm 0.2$°C), which was used for back-up/validation. Water temperature measurements were carried out shading the spring from direct sunlight and avoiding probe contact with sediments, rocks, and vegetation. The calibration of the two instruments was checked at the beginning and at the end of the annual campaigns using an ice bath. In addition, we assessed runoff by a quick visual estimation (always the same operator) similar to Strobl et al. (2020), who considered average width, mean depth and velocity of the flow downslope of the spring. This approach was used to rule out springs with very low runoff (<0.1 l/s).

**3.3 Data analysis**

Before proceeding with statistical analyses, we preliminary filtered field data to exclude problematic or redundant measurements. First, we discarded measurements that were clearly affected by very low runoff (<0.1 l/s), responsible of

large temperature fluctuations during the day (Seppi, 2006). We then selected one measurement site for each rock glacier
and for groups of springs separated less than 10 m from each other. Spring selection was carried out favouring springs
with highest runoff, repeated readings in the four years, closest location to rock glacier fronts, and with lowest interannual
temperature variability.
After this selection, 131 springs were retained. We characterise the springs using different variables (Table 3), namely
the topographic characteristics of the catchments draining to the springs, the activity-state, topographic,
geomorphological, and vegetation characteristics of rock glaciers, and the topographic, geomorphological, geological,
vegetation and permafrost characteristics of the area immediately upslope of the springs. The latter is defined by the
intersection of the catchment perimeter with a circular buffer zone with a radius of 100 m (Fig. 2; Carturan et al., 2016).
Details on these variables, the methodology and the data sources (e.g., DEMs, orthophotos, geological maps and literature)
employed to derive them are listed and described in Table 3.

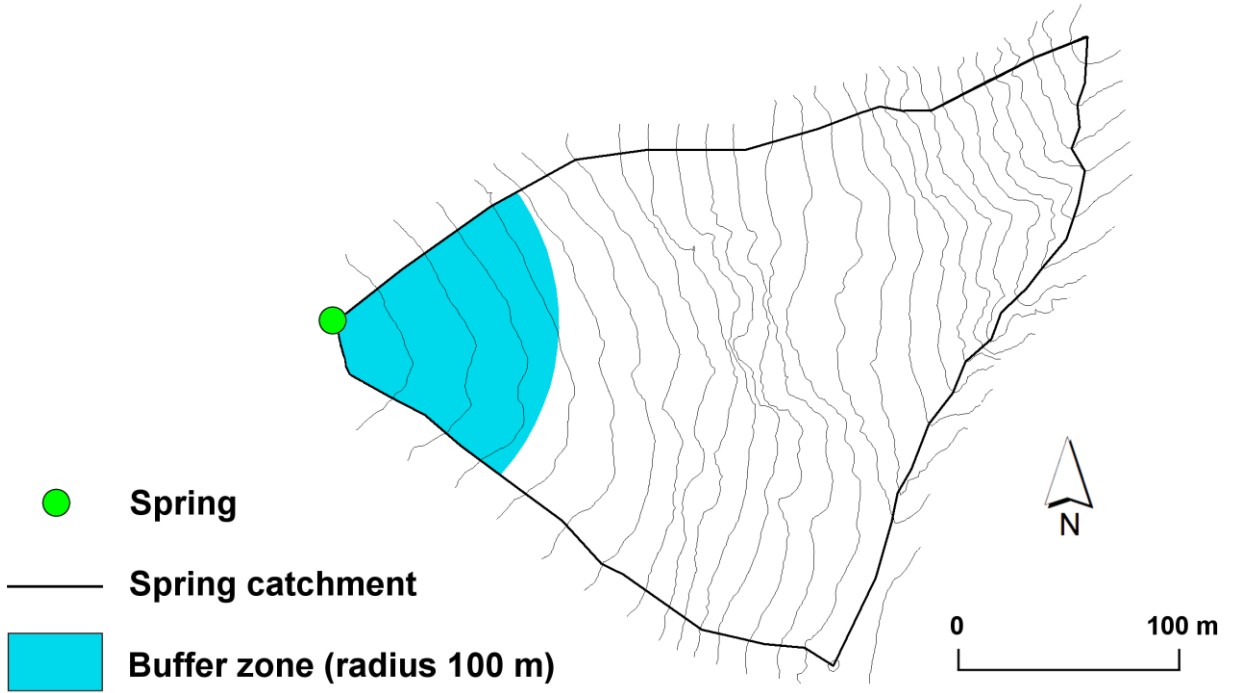


Figure 2: Delimitation of the spring upslope area, defined by the intersection of a circular buffer zone with a radius of
100 m over the catchment perimeter. The methodology was introduced and tested in Carturan et al. (2016).













Table 3: Quantitative and qualitative variables used for characterizing spring areas and for statistical analyses.

| Spatial scale | Variable type | Variable | Classes/acronym | Meaning |
|---|---|---|---|---|
| **Catchment** | **Quantitative** | **Minimum elevation (m a.s.l.)[a]** | \ | Spring elevation |
| | | **Maximum elevation (m a.s.l.)[a]** | \ | |
| | | **Mean elevation (m a.s.l.)[a]** | \ | Half sum of minimum and maximum elevations |
| | | **Planimetric length (m)[a]** | \ | |
| | **Qualitative** | **Mean aspect[a]** | NW-NE | from 315° to 45° |
| | | | NE-SE, SW-NW | from 45° to 135° and from 225° to 315° |
| | | | SE-SW | from 135° to 225° |
| **Spring upslope area** | **Qualitative** | **Geomorphology[b,g]** | ver | Slope deposit (scree slope or debris cone) |
| | | | glac | Glacial deposit |
| | | | rg | Rock glacier |
| | | | pr | Protalus rampart |
| | | | rp | Bedrock |
| | | | df | Debris flow deposit |
| | | | ls | Solifluction lobe |
| | | **Lithology[b]** | TTP | Sillimanite paragneiss (Tonale Unit) |
| | | | TUG | Granate and cyanite paragneiss (Ultimo Unit) |
| | | | TUO | Orthogneiss (Ultimo Unit) |
| | | | OME | Chlorite e sericite micascists (Peio Unit) |
| | | | OMI | Granate and staurolite micascists (Peio Unit) |
| | | | OOG | Orthogneiss (Peio Unit) |
| | | | TPN | Metapegmatites (Tonale Unit) |

| | | | TTM | Marbles (Tonale Unit) |
|---|---|---|---|---|
| | | **Vegetation cover[c]** | 1 | 0-10% covered by vegetation |
| | | | 2 | 10-50% covered by vegetation |
| | | | 3 | 50-90% covered by vegetation |
| | | | 4 | 90-100% covered by vegetation |
| | | **Permafrost evidence[a,c,h]** | weqt | winter equilibrium temperature measured by temperature data loggers |
| | | | geophys | geophysical investigations (this work) |
| | | | snow | perennial snowfields |
| | | | movement | surface displacement visible in multi-temporal DEMs |
| | | | none | no evidence available |
| | | **APIM[d]** | Blue | permafrost in nearly all conditions |
| | | | Purple | permafrost mostly in cold conditions |
| | | | Yellow | permafrost only in very favorable conditions |
| | | | White | no permafrost |
| | | **Open work deposit[e,g]** | Open work | present |
| | | | No open work | absent (includes boulder deposits with fine infill and/or widespread vegetation cover) |
| **Rock glacier** | **Quantitative** | **Front slope (degrees)[a]** | \ | |
| | **Qualitative** | **Activity[f,g]** | Active/transitional | Active/transitional rock glacier |
| | | | Relict | Relict rock glacier |
| | | **Length[a]** | Short | Short rock glacier length class (as defined in Sect. 3) |
| | | | Mid | Mean rock glacier length class (as defined in Sect. 3) |
| | | | Long | Long rock glacier length class (as defined in Sect. 3) |
| | | **Elevation[a]** | Low | Low rock glacier elevation class (as defined in Sect. 3) |
| | | | Mid | Mean rock glacier elevation class (as defined in Sect. 3) |
| | | | High | High rock glacier elevation class (as defined in Sect. 3) |
| | | **Vegetation cover[c]** | Vegetation | Vegetated rock glacier (as defined in Sect. 3, Table 1) |
| | | | No vegetation | Non vegetated rock glacier (as defined in Sect. 3, Table 1) |

| | | Front characteristics[g] | I | No vegetation, evidence of recent instability, outcrop of fine material, little or no surface weathering, weathering degree lower than the surface of the rock glacier |
|---|---|---|---|---|
| | | | II | Very little or no vegetation (<20%), very little or no fine material, weathering and lichen cover comparable to the surface of the rock glacier |
| | | | III | Scarce or discontinuous and cold-adapted vegetation (≤50%), abundant debris, weathering similar to the surface of the rock glacier, cold air draining from voids among blocks |
| | | | IV | Completely vegetated, little outcropping debris, without voids and cold air drainage |
| | | Subdued topography[a,g] | y | The lateral and frontal ridges are clearly evident and the central part of the rock glacier is depressed with respect to them (concave contour lines) |
| | | | n | Lateral ridges are absent or evident only in the upper part of the rock glacier, from halfway down the morphology is convex or almost flat |


[a] Derived from the 2006 and 2014 LiDAR DEM of the Trento Province (siat.provincia.tn.it)
[b] Derived from the 1:10000 geological map of the Trento Province (protezionecivile.tn.it)
[c] Derived from the 2014 orthophoto of the Trento Province (siat.provincia.tn.it)
[d] Derived from the Boeckli et al. (2012) Alpine Permafrost Index Map
[e] Derived from the hillshaded 2014 LiDAR DEM of the Trento Province (siat.provincia.tn.it)
[f] Derived from the Seppi et al. (2012) rock glacier inventory
[g] Derived from field observations
[h] Ground surface temperature data reported in Carturan et al., (2016) and references therein;
http://www.protezionecivile.tn.it/.

We investigated the possible relationship of each variable with the spring-water temperature by means of scatterplots,
boxplots, analysis of variance (or Kruskal-Wallis one way analysis of variance on ranks when variances were not
homogeneous), Dunn's multiple comparison test, Student's t-test, and regression analysis. We defined spring-water
temperature as "the median of all available temperature measurements in the four years", so that we smoothed the
interannual variability of water temperature. However, we had also to account for the different number of measurements
available for each spring (from one up to four), and in particular for the possible low representativeness of springs
measured only once. In this case, there is the possibility of having measured an extreme value, far from the typical
conditions of those springs. To evaluate the impact of extreme values, we computed the absolute difference between each
single-year spring water measurement and the median of all available measurements at the same spring. The mean of
these absolute differences was 0.12°C, the median was 0.05°C, whereas the minimum and the maximum were 0 and
0.7°C, respectively, and 89% of values was below 0.3°C. These results indicate a low impact of extreme temperatures
and the suitability of using the median of all available measurements (regardless of their number) in statistical analyses.
For springs with temperature measured only once, we retained the single value if runoff was >0.1 l/s.

**3.4 Geophysical investigations**

Electrical resistivity tomography (ERT) surveys were performed on 13-14 July 2022 at two neighbouring rock glaciers,
classified as relict in the inventory of Seppi et al. (2012), to constrain spring-water temperature results at local scale.
These rock glaciers were selected considering their different characteristics (spring-water temperature, vegetation cover,
elevation) and the easy access. In addition, they have uniform lithology, which minimises the uncertainty in the
interpretation of gradients in electrical resistivity. The Preghena Rock Glacier has a mean elevation of 2196 m a.s.l., is
mainly free of vegetation (although shrubs and trees are present) and its spring-water temperature ranged between 1.6 and
1.8°C throughout the late summer during the measuring period. The Bordolona Rock Glacier has a mean elevation of
1967 m a.s.l., is completely covered by vegetation and its spring-water temperature ranged between 3.5 and 3.7°C in the
late summer during the measuring period. Both rock glaciers are northeast oriented (Fig. 3).

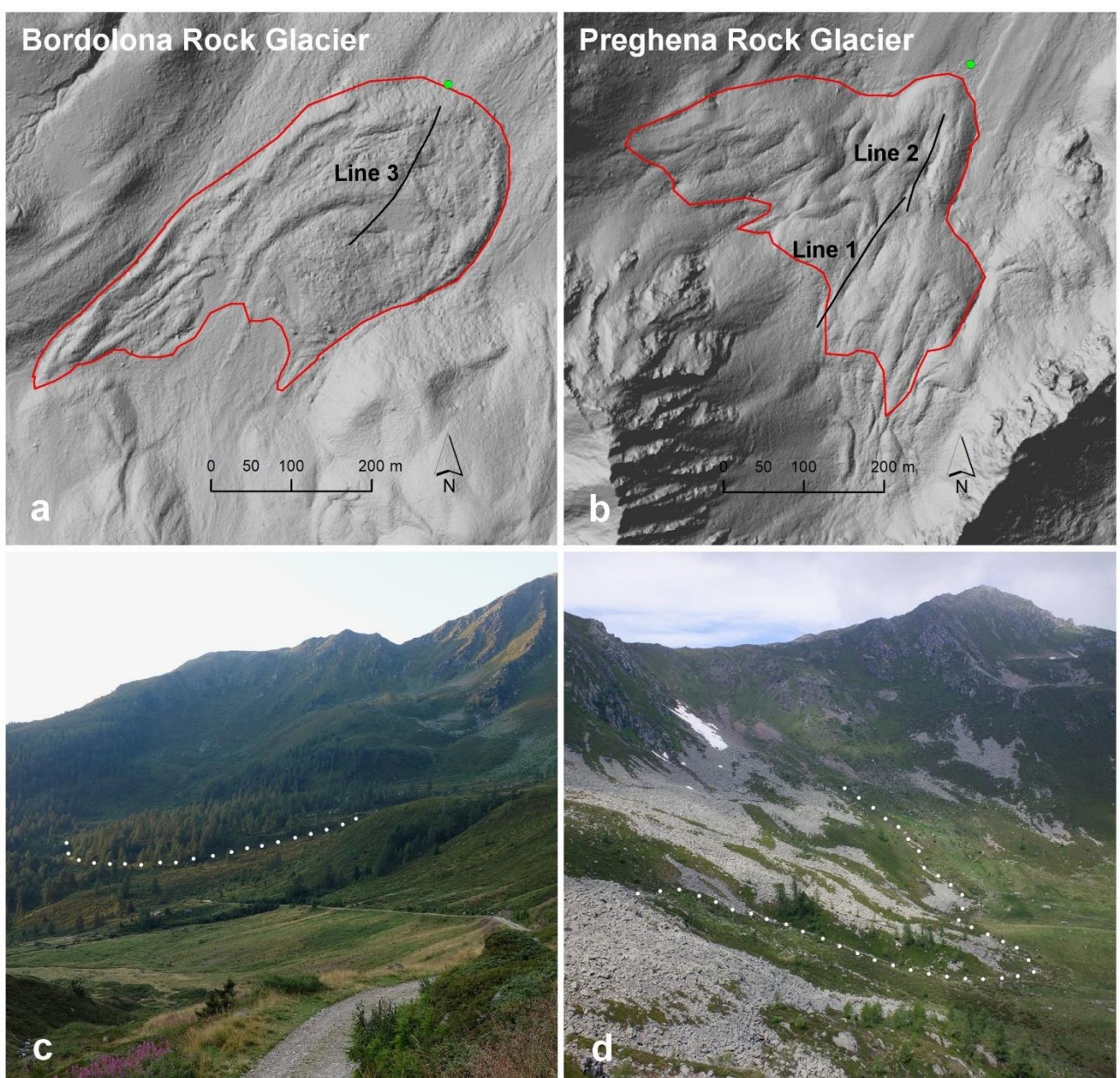

Figure 3: Location of ERT lines (black solid lines) performed on the a) Bordolona and b) Preghena rock glaciers in July 2022. The green dots in a) and b) indicate the sampled springs. The white dots in c) and d) indicate the lower edge of Bordolona and Preghena rock glaciers, respectively.

Geophysical surveys were carried out with a Syscal Pro georesistivimeter (Iris Instruments), using arrays of 72 (Line 1 Preghena and Line 3 Bordolona) or 48 (Line 2 Preghena) electrodes, with 3-meter electrodes spacing (Fig. 3). A total length of 346 and 216 m was investigated at the Preghena and Bordolona rock glaciers, respectively. A dipole-dipole scheme was used, with two different skips of 0 and 4 electrodes. This configuration ensured relatively high resolution at the surface, and at the same time enough penetration depth. Measurements were carried out with a stack of 3 to 6, imposing an acceptable error threshold of 5%. To estimate a more reliable experimental error for the acquired datasets (Binley, 2015), direct and reciprocal measurements were acquired by exchanging injecting and potential dipoles for each quadrupole. To partially overcome the high contact resistances between the electrodes and boulders/debris (Hauck and

Kneisell, 2008), the electrodes were inserted between the boulders using sponges soaked with saltwater (Pavoni et al.,
2023). Nevertheless, at the blocky surface of the Preghena Rock Glacier the contact resistances remained steadily above
$10^5$ Ωm, due to dry environmental conditions. The organic soil at the Bordolona Rock Glacier guaranteed low contact
resistances (<$10^4$ Ωm).
The inversion process of the acquired datasets has been performed with the Python-based software ResIPy (Blanchy et
al., 2020), based on the Occam's inversion method (Binley and Kemna, 2005). In each dataset, quadrupoles with a
stacking error higher than 5% were removed, and the expected data error was defined using the reciprocal check (Day-
Lewis et al., 2008, Pavoni et al., 2023), giving values of 20% and 5% for the Preghena and Bordolona Rock Glacier,
respectively.

**274  3.5 Calculation of ice storage in the rock glaciers and glaciers**

In order to estimate and compare the ice content of rock glaciers and glaciers in Val di Sole, we applied an approach
similar to the one used by Bolch and Marchenko (2009) in the Northern Tien Shan. For glaciers, we estimated residual
volumes in 2022 starting from the 2003 ice thickness estimates provided for each glacier in the study area by Farinotti et
al., (2019). We first calculated the bedrock topography subtracting the ice thickness from the glacier surface DEM
(Farinotti et al., 2019). Then we calculated the 2022 glacier thickness subtracting the bedrock topography from a glacier
surface DEM surveyed in September 2022 by the Province of Trento. We finally obtained the glacier volumes multiplying
the average thickness by the glacier area, and converted the ice volume into the water volume equivalent using a mean
ice density of 900 kg m$^{-3}$.
For rock glaciers, we calculated the total rock glacier volume multiplying their area $A$ by the average thickness provided
by the Brenning (2005b) formulation:
$$T = cA^\gamma \tag{1}$$
where $T$ is the average thickness of rock glaciers, and $c$ and $\gamma$ are constants equal to 50 and 0.2, respectively. To account
for the different geometry of active/transitional and relict rock glaciers, we assumed that the volumetric ice content of
active/transitional rock glaciers averages 50% (Jones et al., 2018, and references therein), and therefore that $T_r$ for (true)
relict rock glaciers is half that of active/transitional rock glaciers (i.e., they are composed only of debris and all the ice
melted away). For pseudo-relict rock glaciers we tested various hypotheses of percent ice content, ranging between 5%
and 20%, calculating the average thickness $T_{pr}$ as follows:
$$T_{pr} = T_r + T_{ice} \tag{2}$$
where $T_{ice}$ is the average ice thickness, calculated in function of the volumetric percent ice content %$_{ice}$ as:
$$T_{ice} = \frac{\%_{ice} \cdot T_r}{(1 - \%_{ice})} \tag{3}$$

**296  4. Results**

**297  4.1 Spatial variability of spring-water temperature**

Water temperature of the 131 springs ranged between 0.0 and 8.5 °C, with a mean of 3.6 °C and a median of 3.4°C (Table 4). The frequency distribution of the spring elevation (i.e., the minimum elevation of catchments) is symmetrical and normally distributed around a sample mean of 2384 m a.s.l. The lowermost spring was sampled at 1698 m a.s.l., and the uppermost spring was sampled at 3039 m a.s.l.

The mean elevation of spring catchments varies between 2104 and 3151 m a.s.l., whereas the maximum elevation ranges between 2241 and 3352 m a.s.l. The mean and maximum elevation average 2539 and 2694 m a.s.l., respectively. Both are also symmetrical around the sample mean and normally distributed.

The planimetric length of spring catchments varies between 83 and 2621 m, with a mean of 610 m. The skewness and kurtosis indicate that the planimetric length is right skewed and leptokurtic.

Table 4: Descriptive statistics for spring-water temperature measurements and quantitative variables relative to spring catchments (as defined in Table 3).

| N = 131 | Median temperature ($T_{Mdn}$) | Catchment minimum elevation (m) | Catchment maximum elevation (m) | Catchment mean elevation (m) | Catchment planimetric length (m) |
|---|---|---|---|---|---|
| Minimum | 0.0 | 1698 | 2241 | 2104 | 83 |
| Median | 3.4 | 2367 | 2641 | 2495 | 539 |
| Maximum | 8.5 | 3039 | 3352 | 3151 | 2621 |
| Range | 8.5 | 1341 | 1111 | 1047 | 2538 |
| Mean | 3.6 | 2384 | 2694 | 2539 | 610 |
| Standard error of the mean | 0.2 | 22.6 | 21.9 | 21.0 | 34.9 |
| Standard deviation | 1.8 | 259.2 | 251.1 | 240.8 | 399.3 |
| Coefficient of variation | 0.500 | 0.109 | 0.093 | 0.095 | 0.655 |
| Skewness | 0.392 | 0.179 | 0.446 | 0.419 | 2.070 |
| Kurtosis | -0.261 | -0.328 | -0.107 | -0.391 | 6.095 |

Spring-water temperature is significantly correlated with the mean elevation of the catchments (Fig. 4a) for all three aspect classes defined in Table 3. Linear regressions are significant (p< 0.001) for south ($R^2 = 0.30$) and for east-west facing catchments ($R^2 = 0.35$). For the north facing catchments, there is a low significant relation ($R^2 = 0.25$, p<0.05) between water temperature and elevation. In all three cases, the low $R^2$ suggests that other factors should affect water temperature, as well. Similar results were obtained using spring elevation rather than mean catchment elevation (Fig. 5).

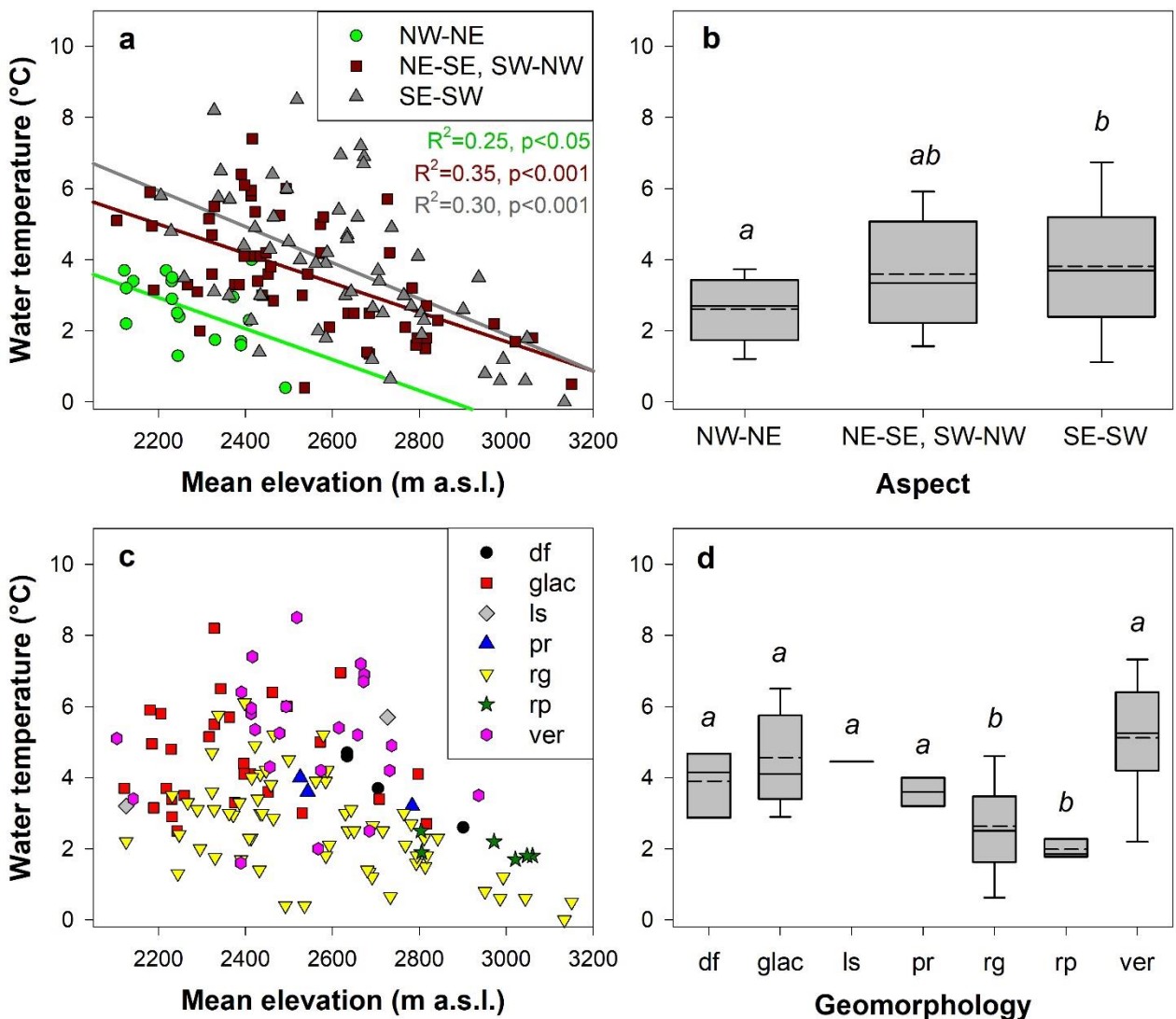


Figure 4: Relationship between spring-water temperature and a) mean catchment elevation (clustered in three classes of
mean catchment aspect), b) mean catchment aspect, c) mean catchment elevation (clustered in seven classes of upslope
area geomorphology), and d) upslope area geomorphology. Acronyms and their meanings are reported in Table 3. Boxes
in b) and d) indicate the 25th and 75th percentile, whiskers indicate the 10th and 90th percentile, whereas the horizontal
solid and dashed lines within the box mark the median and the mean, respectively. Different letters above the boxplots
indicate groups with significantly different (p<0.05) water temperatures based on Dunn's multiple comparison test
(applied after the Kruskal-Wallis test).




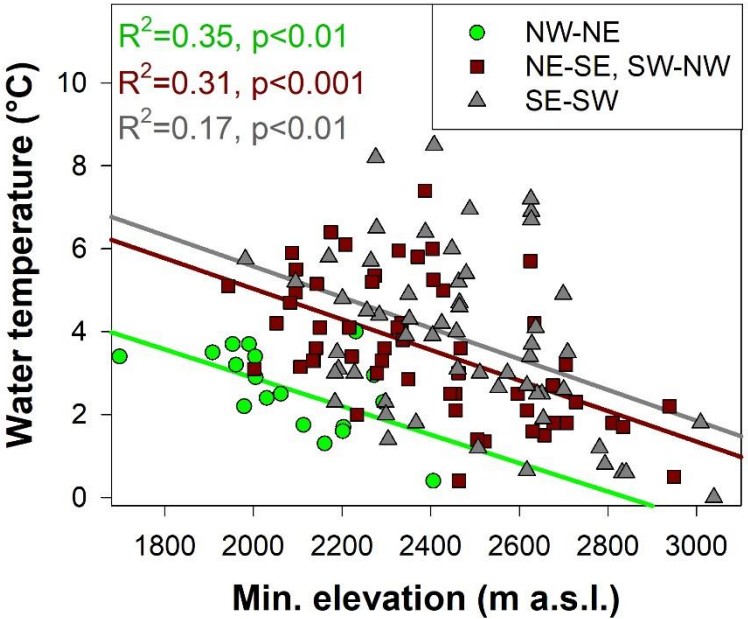


Figure 5: Relationship between spring-water temperature and minimum (spring) elevation, clustered in three classes of mean catchment aspect, as in Fig. 4a.

As expected, there is a negative relationship between water temperature and elevation (Fig. 4a and 5), but also a large overlap of water temperature among the three aspect classes. NW-NE facing catchments have significantly colder springs compared to SE-SW facing catchments (p<0.05, Dunn's multiple comparison test, applied after the Kruskal-Wallis test), whereas catchments facing NE-SE and SW-NW have water temperature that do not differ significantly from the other two classes (Fig. 4b). NW-NE facing catchments show a lower variability in spring-water temperature compared to the other two classes.

Figures 4c and 4d highlight that springs with upslope areas dominated by the presence of rock glaciers (irrespective of their activity) and bedrock outcrops are significantly colder than other springs (p<0.05, Dunn's multiple comparison test, applied after the Kruskal-Wallis test).

**4.2 Temperature of springs downslope of rock glaciers**

**4.2.1 Comparison between active/transitional and relict rock glaciers**

The spring-water temperature is significantly different for rock glaciers with different degrees of activity (Fig. 6a). Relict rock glaciers have a much warmer spring temperature compared to active/transitional rock glaciers (Student's t-test, p<0.001), and the variability of water temperature is larger for relict rock glaciers. There is a substantial overlap between the two groups, which extended between 1.2 and 3.0°C. This range of water temperature represents 54% of all springs downslope of rock glaciers (53% of active/transitional rock glaciers and 54% of relict rock glaciers). Almost half of rock glaciers classified as relict has spring-water temperature similar to rock glaciers classified as active/transitional.

The two groups of rock glaciers have significantly different minimum elevations (Fig. 6b, Student's t-test, p<0.001), but there is a wide elevation band, comprised between 2406 and 2630 m a.s.l., where they overlap.

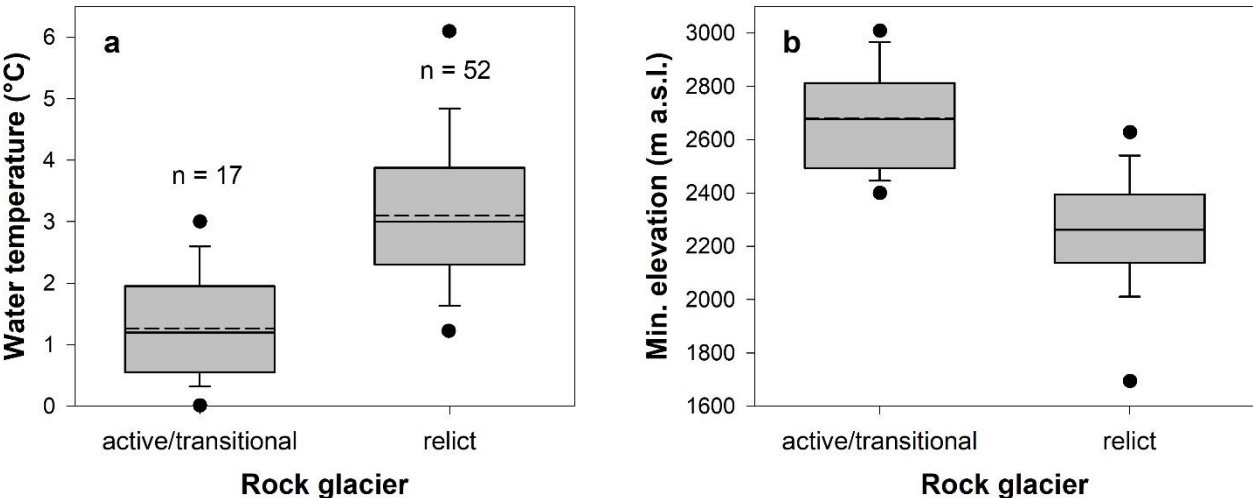

Figure 6: Spring-water temperature (a) and minimum elevation (b) of rock glaciers sampled in the study area. Boxes indicate the 25th and 75th percentile, whiskers indicate the 10th and 90th percentile, whereas the horizontal solid and dashed lines within the box mark the median and the mean, respectively. Maximum and minimum values are represented by dots. Sample size (n) is reported in a).

**4.2.2 Spring-water temperature of relict rock glaciers**

The relationship between water temperature and the mean catchment elevation is rather weak for springs fed by relict rock glaciers (Fig. 7a). The linear regression is significant ($p<0.05$) only for catchments facing NE-SE and SW-NW, but the relation is weak ($R^2 = 0.20$). At the same elevation, catchments facing NW-NE have colder springs compared to the other two aspect classes. The spring-water temperature of catchments facing north is similar to that of catchments facing east, south and west located 300-400 m above.

Relict rock glacier springs with open work deposits in their upslope areas are colder than springs without open work deposits (Fig. 7b). For the first group, the water temperature is not related to the mean catchment elevation, whereas for the second group there is a weak but significant relation ($p<0.05$, $R^2 = 0.15$). Consequently, the difference in water temperature of the two groups increase towards low elevations, which suggest that open work deposits may have a cooling effect particularly marked at elevations <2500 m a.s.l.

Similar considerations can be done for rock glacier front characteristics (Fig. 7c) and for rock glacier vegetation cover (Fig. 7d). Relict rock glaciers with scarce and cold-adapted vegetation cover have colder springs compared to relict rock glaciers with abundant vegetation cover on their bodies and fronts. However, for all classes of rock-glacier front characteristics and vegetation cover (Table 3) there is no significant relation between water temperature and mean catchment elevation.

Despite the large overlap among the analysed classes (Fig. 7), we found a significant effect of vegetation cover (Student's t-test, $p<0.001$), open work deposits (Student's t-test, $p<0.001$) and front characteristics (Student's t-test applied to classes III and IV, $p<0.01$) on the water temperature of springs downslope of relict rock glaciers. We did not detect any significant influence of the mean aspect of the catchment, the mean elevation of rock glaciers, their length, and the presence or absence of a subdued topography on water temperature.

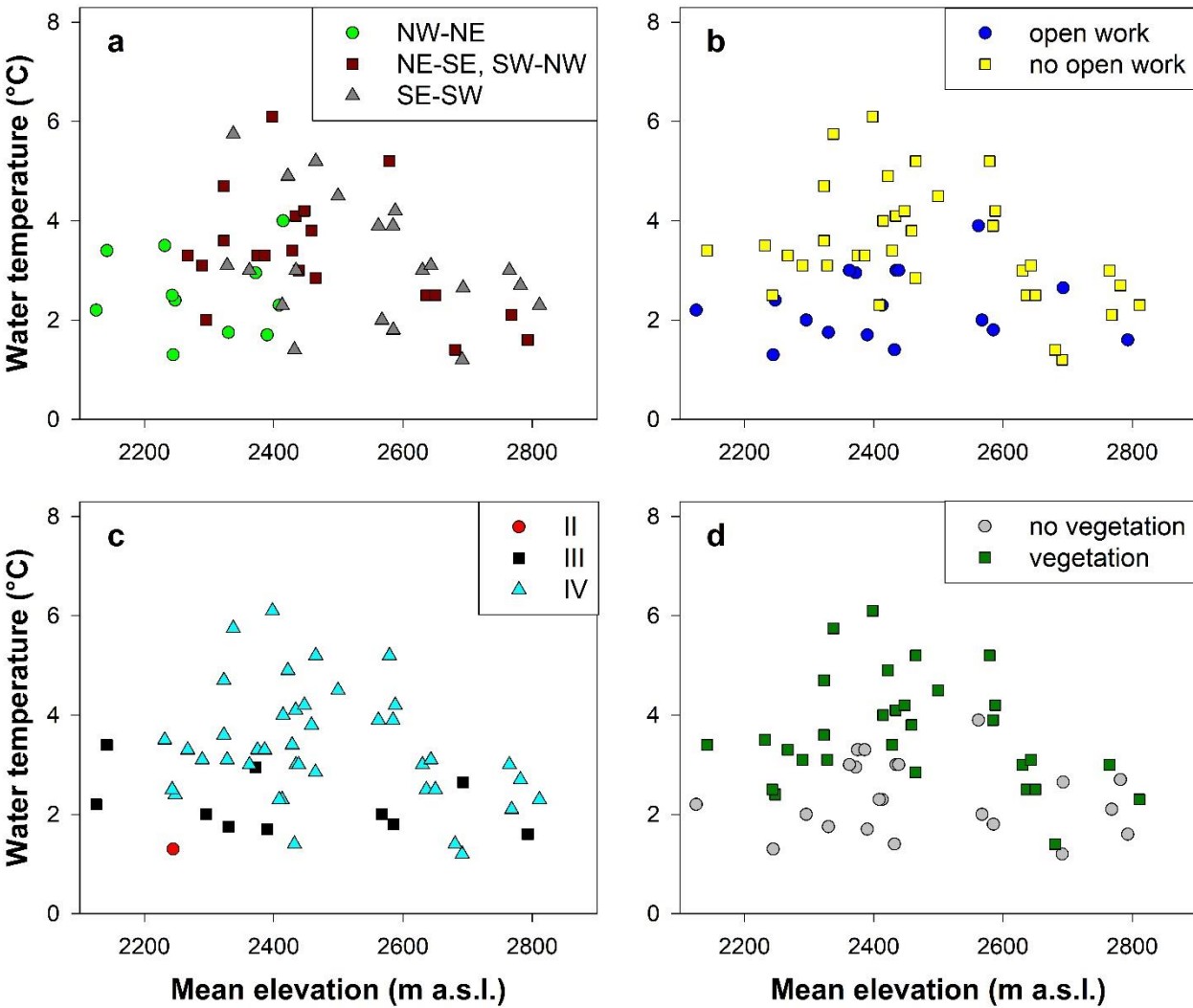


Figure 7: Relationship between spring-water temperature of relict rock glaciers and mean catchment elevation clustered
in a) three classes of mean catchment aspect, b) two classes of open work deposits in the spring upslope area, c) three
classes of rock glacier front characteristics, and d) two classes of rock glacier vegetation cover. Classes are described in
Table 3.

**4.3 Geophysical investigations**
Figures 8a and b show the inverted resistivity sections obtained for the investigation Lines 1 and 2 acquired on the
Preghena Rock Glacier. High values of resistivity ($>8 \cdot 10^4$ $\Omega$m) were found in the uppermost layer, down to about 7-8
meters of depth, associated to the dry conditions during ERT soundings and to the air-filled voids among coarse debris
and blocks, typical of rock glacier environments. Below this uppermost layer, the resistivity values rapidly decrease ($<10^4$
$\Omega$m) indicating a plausible decrease of porosity and grain size in the deposit, and a possible increase in water content.
This low resistivity layer develops almost continuously down to the bottom of the models. An increase in resistivity is
found at the lower end of line 1 and at the upper end of line 2, in the area where they overlap and at a depth of about 12-
13 m, reaching 1.5-2.0·$10^5$ Ωm. This area of increased resistivity can be interpreted as a deep frozen body, providing
evidence of probable permafrost inside this rock glacier.
Figure 8c shows the inverted resistivity section obtained for the investigation Line 3 acquired on the Bordolona Rock
Glacier. In the shallowest layers the resistivity is comprised between 5·$10^3$ and $10^4$ Ωm, significantly lower than the
shallow layer of the Preghena Rock Glacier, even if air-filled voids are common on this rock glacier as well.
Below this layer, a sharp increase in resistivity is detected along the entire investigation line, with frequent regions
exceeding 2·$10^4$ Ωm. The highest resistivity (about 6·$10^4$ Ωm) is found towards the upper end of the ERT line, where a
younger rock glacier lobe overlies the main body. This high resistivity layer reaches about 15 meters of depth and can be
interpreted as a frozen layer. The bottom of the high-resistivity layer, which seems discontinuous in the lower part and
more continuous and thicker in the upper part of the ERT line, is highlighted by a strong decrease in resistivity, below
5·$10^3$ Ωm. This lowermost layer is probably unfrozen and is characterised by an increase in water content and fine
sediments.

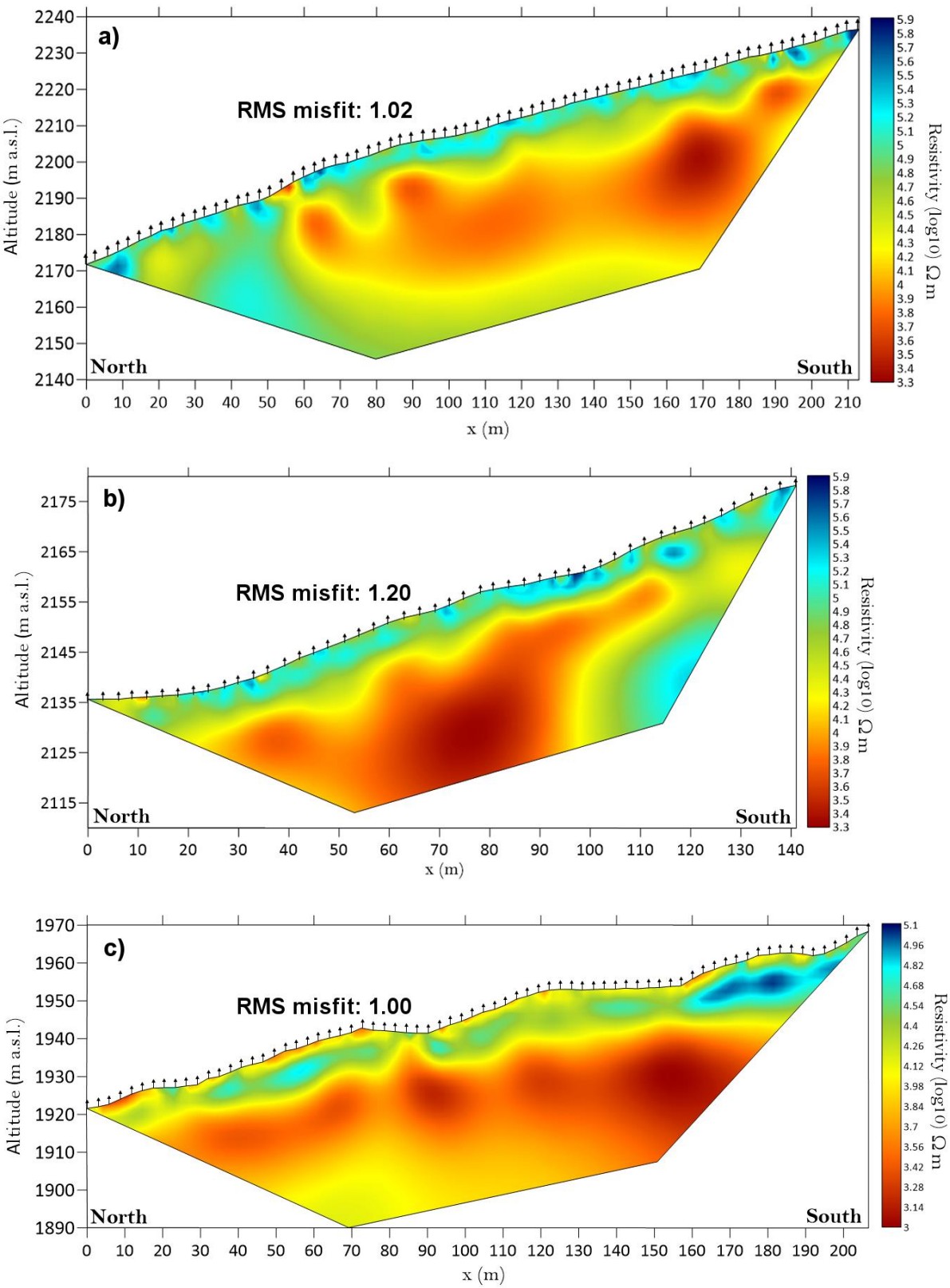

Figure 8: Inverted resistivity section of the investigation Line 1 (a) and 2 (b) on the Preghena Rock Glacier, and of the investigation line 3 (c) on the Bordolona Rock Glacier.

**4.4 Ice storage in the rock glaciers and glaciers**

A total glacier ice volume of 251 x $10^6$ m$^3$, and a corresponding 226 x $10^6$ m$^3$ water volume equivalent was calculated for
Val di Sole in 2022. In comparison, the water volume equivalent of active/transitional rock glaciers is 42.7 x $10^6$ m$^3$.
A water volume equivalent between 4.4 and 20.9 x $10^6$ m$^3$, averaging 12.7 x $10^6$ m$^3$, can be estimated assuming that
50% of the total area of relict rock glaciers contains permafrost (rounded value, based on results reported in Section
4.2.1), and that the average ice content ranges between 5% and 20% in volume.

**5. Discussion**
**5.1 Permafrost distribution and spring-water temperature in the study area**
Measurements of spring-water temperature collected in this study outside the rock-glacier influence have a high spatial
variability and do not show a significant relationship with elevation (p>0.05). Among springs outside the rock glacier
influence, only those above 2800 m a.s.l. have a water temperature ≤2.2°C, which is the upper limit reported in the
literature for 'possible permafrost' (Carturan et al., 2016).
This result lines up well with mean annual air temperature (MAAT) indications. Indeed, based on the MAAT of -0.9°C
measured between 1961 and 2010 at the Careser Diga weather station (2605 m a.s.l., in the northern part of the Val di
Sole), the theoretical lower limit of discontinuous permafrost in Val di Sole, corresponding to a MAAT of -2°C (Haeberli,
1985), should be comprised between 2700 and 2800 m a.s.l..
Similarly, the alpine permafrost index map (APIM, Boeckli et al., 2012) indicates a lower limit of "permafrost mostly in
cold conditions" ranging between 2500 and 2900 m outside rock glaciers and coarse-block deposits, varying upon terrain
aspect and averaging 2700 m a.s.l.. Based on the mean elevation of active/transitional rock glaciers in the study area,
Seppi et al. (2012) calculated a present-day lower limit of permafrost at 2720 m a.s.l..
As expected, springs draining north-facing catchments are significantly colder compared to springs draining south-facing
catchments. On average, there is a difference of about 3°C between springs draining catchments at similar elevation and
with opposite aspect. On average, the same spring temperature is found 500-600 m higher on south-facing catchments
than on north-facing ones (Fig. 5). This result quantifies the influence of terrain exposure on the ground temperature
regime and permafrost distribution in the study area, which are direct consequences of shortwave radiation inputs and
related effects on snow cover and surface albedo (Boeckli et al., 2012; Wagner et al., 2019; Amschwand et al., 2024).
In our study, at all elevations, springs draining rock glaciers are the coldest, irrespective of the rock glacier activity state
(Fig. 4c). This is in agreement with findings of studies in the European Alps and in other mountain chains reporting rock-
glacier spring-water temperatures, regardless of their activity state. For example, in the Canadian Rockies, spring-water
temperature from an inactive rock glacier hosting small portions of permafrost reached a maximum of 2.2 °C, exercising
a substantial cooling effect on the creek downstream (Harrington et al., 2018). Interestingly, cold conditions and high
daily variability in spring-water temperature in summertime has been recorded in a rock glacier in Norway that shows
characteristics favourable to the presence of permafrost, but with minor ice bodies (Lilleøren et al., 2022). In the Austrian
Alps, spring-water from a relict rock glacier was monitored for 6 years, showing a mean temperature of 2.2°C, with small
seasonal variation (between 1.9 and 2.5 °C) and a decrease of the water temperature after precipitation events, attributed
to the potential presence of ice lenses in the lower part of the rock glacier (Winkler et al., 2016).
Our results align as well with those of studies reconstructing permafrost distribution by empirical modelling in the Alps
and at other mountain locations worldwide. A logistic regression model used in the Dry Andes of Argentina accounting
for mean annual air temperature, terrain ruggedness, and potential incoming solar radiation suggests that permafrost may
occur in several types of coarse blocky deposits, including rock glaciers, even under unfavourable climatic conditions
(Tapia Baldis and Trombotto-Liaudat, 2020). A similar empirical-statistical model applied in the Austrian Alps shows
that permafrost can be expected above 2500 m a.s.l. in northerly exposed slopes and above 3000 m a.s.l. in southerly
exposed slopes (Schrott et al., 2012), providing an elevation difference of about 500 m between south and north exposures,
which agrees well with our spring-water temperature results.

**5.2 Rock glacier classification based on spring-water temperature**

Although springs draining active/transitional rock glaciers are significantly colder than springs draining relict rock
glaciers, there is a remarkable ~50% overlap in the water temperature range of the two rock glacier groups (Fig. 6a).
Based on published thresholds (Haeberli, 1975; Frauenfelder et al., 1998; Scapozza, 2009, Carturan et al., 2016), 12 out
of the 52 relict rock glaciers sampled in Val di Sole (23%) can be included in the 'possible permafrost' category (water
temperature between 1±0.2 and 2±0.2 °C), and none of them in the 'probable permafrost' category (water temperature <
1±0.2 °C).  However, the relatively warm water temperature measured downstream of active/transitional rock glaciers
(maximum = 3°C, 90[th] percentile = 2.4°C), and downstream of areas with permafrost evidence (maximum = 3.5°C, 90[th]
percentile = 2.2°C), suggest that the upper limit of spring-water temperature for possible permafrost may be higher. Here,
the 90[th] percentile accounts for possible misclassification of active/transitional rock glaciers and other issues affecting
spring-water temperature measurements (Sect. 5.3).
Assuming a (rounded) upper limit of 2.5°C for spring-water temperature with possible permafrost influence leads to
include 19 (38%) relict rock glaciers in the possible permafrost category. This estimate looks more conservative than the
~50% obtained by a mere comparison of water temperature ranges of active/transitional and relict rock glaciers (Fig. 6a).
These findings might suggest that permafrost in rock glaciers classified as relict is widespread in Val di Sole, and that a
large fraction of them is actually pseudo-relict, or transitional landforms, containing patches of permafrost and reaching
an elevation below the tree line (2000-2200 m a.s.l.). Compared to the rock glacier classification performed by Seppi et
al. (2012), which was based on remote-sensing geomorphometric evidence combined to field observations (topographic
surveys and ground surface temperature measurements for few rock glaciers), spring-water temperature suggests the need
for a reclassification of a large fraction of rock glaciers categorised as relict into pseudo-relict.
Examples of spring-water temperature downstream of rock glaciers in Val di Sole are shown in Fig. 9. Cold springs
draining rock glaciers classified as relict are associated to the presence of open work deposits and scarce vegetation cover
(Fig. 7 and 9). These two explaining variables are often correlated, because vegetation tend to be scarce over coarse
deposits without fine infill among blocks, and vice versa. The relationship between cold spring temperature (as permafrost
evidence) and these two surface characteristics was expected in our case study, based on the existing literature (e.g.,
Guglielmin, 1997, and references therein). This relationship is statistically significant only for rock glaciers classified as
relict, whereas for active/transitional rock glaciers sampled in the study area it does not exist (Fig. A1).

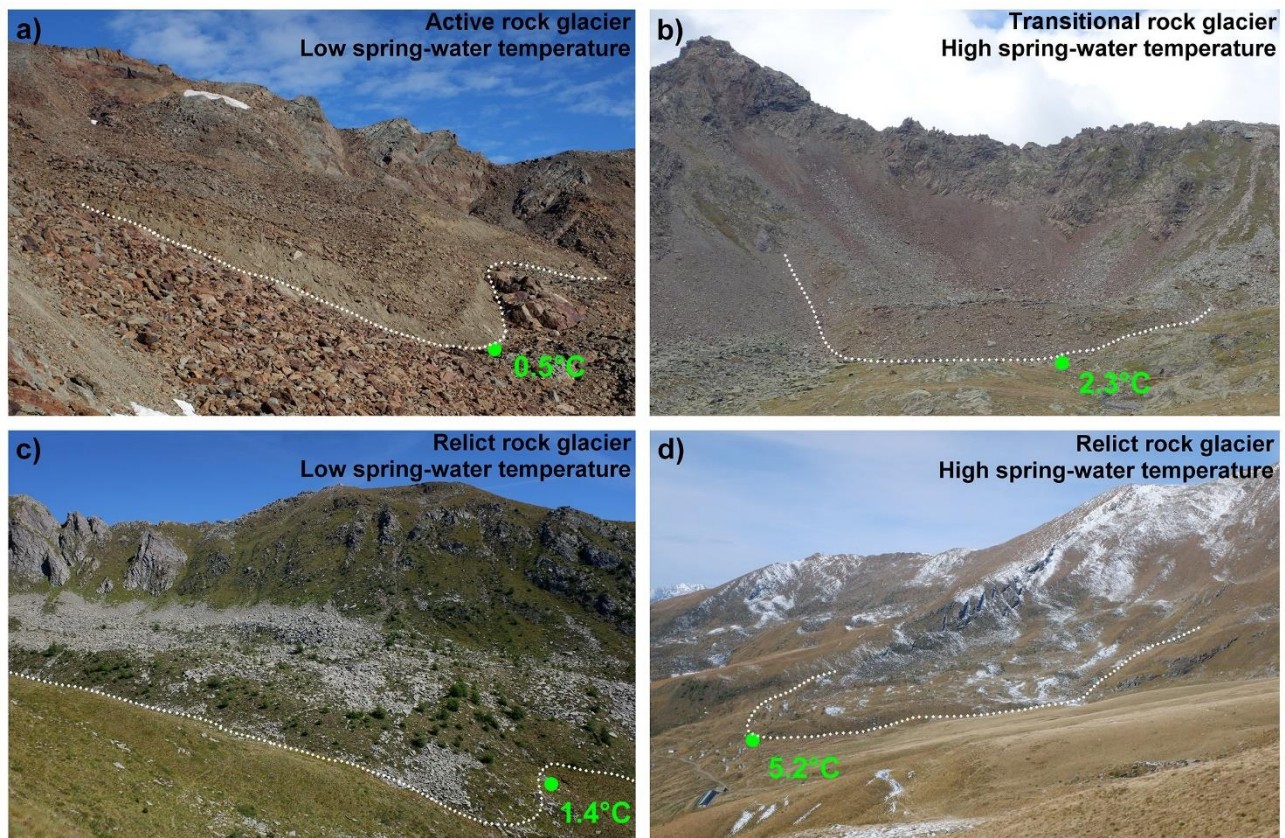


Figure 9: Examples of spring-water temperature downstream of rock glaciers in Val di Sole: a) active rock glacier with cold spring at 2950 m a.s.l.; b) transitional rock glacier with relatively warm spring at 2727 m a.s.l.; c) relict rock glacier with cold spring at 2304 m a.s.l., whose surface is open-work and presents scarce vegetation cover; d) relict rock glacier with warm spring at 2266 m a.s.l., whose surface is entirely covered by vegetation.

The long-term preservation of permafrost within open work blocky deposits results from overcooling and thermal decoupling of the frozen core from the external climate (Harris and Pedersen, 1998; Morard et al., 2008; Jones et al., 2019). The low thermal conductivity in coarse open work deposits brings to lower ground temperatures compared to fine-grain material (Juliussen and Humlum, 2008; Jones et al., 2019). Soil development over the surficial blocks and boulders can prevent these cooling effects (Ikeda and Matsuoka, 2002). However, if fine-grain infilling does not occur, ground cooling effect goes undisturbed. In central Europe, these processes enable the existence of permafrost much below its regional limit and reaching elevations lower than 1000 m a.s.l. (Gude et al., 2003; Delaloye et al., 2003). According to Delaloye and Lambiel (2005), thousand-year-old permafrost might be potentially preserved in these types of deposits.

Open work deposits and/or scarce vegetation cover can be potentially employed to distinguish rock glaciers with or without permafrost, as both can be mapped based on remote-sensing imagery. However, open work deposits and vegetation cover do not enable a full distinction of 'cold' and 'warm' springs affected by relict rock glaciers (Fig. 7b, c and d). Individual non-open-work rock glaciers widely covered by vegetation can have spring-water temperature as low as 1.4°, and rock glaciers almost free of vegetation with blocky surface can have spring-water temperature up to 3.9°C.

Other variables considered in this study, such as aspect, elevation, size and the presence or absence of a subdued topography on rock glaciers (Delaloye et al., 2003; Delaloye, 2004), are not related to spring-water temperature. Figure 7 suggests the existence of a group of cold springs at low elevations on north-facing catchments, even though water

temperature is not significantly different from the temperatures of springs in the other two aspect classes. This result
might be due to the small sample size of the NW-NE aspect class.

**5.3 Limitations and uncertainties in the spring-water temperature approach**
The results of this study might be affected by limitations in the experimental design, assumptions, and uncertainties. First,
the main assumption of this study is that spring-water temperature provides indication of permafrost occurrence at
investigated rock glaciers and spring upslope areas, and can be used as a stand-alone pilot method to rapidly explore the
activity state of rock glaciers in a wide area. This approach applies spring-water temperature to the catchment scale,
beyond its general use as an ancillary method to other techniques such as InSAR analyses, ground surface temperature
measurements and/or geophysics.
We base our assumption on previously published work and well-known temperature thresholds for permafrost probability
categories (e.g., Haeberli, 1975; Frauenfelder et al., 1998; Scapozza, 2009) and on our first successful application at the
catchment scale (Carturan et al., 2016). Data collected in Val di Sole are in line with literature thresholds, provided that
the 10% largest spring-water temperature values are excluded (Sect. 5.2). Including these extreme values leads to about
1.5°C larger temperature thresholds for possible permafrost compared to literature.
The reason behind this discrepancy lies in the uncertainty in the classification of rock glacier activity, which was based
on vegetation and geomorphological characteristics, assessed mainly from remote-sensing images (Seppi et al., 2012). In
the wide elevation band where active/transitional and relict rock glaciers coexist (minimum elevation between 2406 and
2630 m), landforms with similar vegetation cover and surface geomorphology have been classified based on the authors'
experience and judgement, implying a certain degree of subjectivity.
The distinction between active/transitional and relict rock glaciers is a theoretical concept, and there is a continuum
between transitional and (true) relict rock glaciers (Kääb, 2013). In absence of other evidence, this continuum hampers to
distinguish unambiguously transitional and relict landforms, in particular if they have similar surface characteristics. In
addition, the mentioned transition is a dynamic concept, which depends on the characteristics of individual landforms,
their topo-climatic setting, and their response to climatic variations (Kääb, 2013).
Another source of uncertainty is related to the distance between the permafrost body and the measured springs. Water
temperature is a non-conservative tracer, and if the main permafrost body is distant (e.g., more than one hundred meters)
from the rock glacier front, or if permafrost is patchy and not in contact with groundwater paths, water temperature can
be largely influenced by unfrozen sediments and/or mixing with other water sources (e.g., Kellerer-Pirklbauer et al.,
2017). This is the case of the Bordolona Rock Glacier (Fig. 8c), where the rather warm spring-water temperature (3.5-
3.7°C) would have led to exclude the occurrence of permafrost in absence of geophysical evidence.
For smaller distances, we have checked the impact of spring location downstream of rock glacier fronts at three
measurement sites, where the same stream emerged briefly at the rock glacier front and a few tens meters downstream.
Measurements confirmed that there was negligible warming (from 0.0 to 0.1°C) of the water downstream of the rock
glacier front, at least as long as the water remained below the surface.
Seasonal ice formed in the topmost ground layer during winter and spring, in areas without permafrost, might cool down
spring-water temperature leading to false-positives in permafrost detection. We think that making measurements in late
summer, as proposed by the literature (e.g. Haeberli, 1975), prevents this seasonal ice from affecting spring-water
temperature measurements, or at least strongly minimizes its effect. Possible influence from seasonal ground ice formation
should be largest after cool/short summer seasons, but this was not the case in the study period.
Depending on the measurement time, which was between 8.00 and 18.00 CET, any variation of temperature during the
day might also influence the results. Hourly records of spring-water temperature collected by Seppi (2006) lead us to
exclude significant variation of spring-water temperature during the day, at least for springs with runoff higher than 0.1
l/s.
Several authors are cautious when discussing about cold springs downslope of relict rock glaciers. For example, Winkler
et al. (2016) do not exclude the presence of remaining ice lenses inside the relict Schöneben Rock Glacier (Niedere Tauern
Range, Austria), as a possible explanation for the rapid cooling of the spring water after recharge events, during
summertime. However, the authors mention the cold thermal regime beneath coarse blocky materials as a possible
explanation, which does not necessarily imply permafrost occurrence, and conclude that additional research is required
for the identification of the cooling source.
We agree that additional research is required to confirm inference from spring-water temperature. With this study we add
that spring-water temperature can be as high as 1.8°C for rock glaciers where permafrost occurrence is confirmed by
geophysics or ground surface temperature measurements, and can exceed 3.5°C where the permafrost body is far from
the rock glacier front and spring, such as at the Bordolona Rock Glacier. Even if the collected data seem to suggest that
temperature thresholds might be slightly higher than those reported in the literature, further investigations are necessary
for better constraining them and for defining their range of uncertainty. Based on the evidence discussed in this section,
a warm bias might prevail over a possible cold bias in our spring water data, leading to false-negatives in permafrost
detection. For this reason, the frequency of pseudo-relict reported in Section 5.2 can be considered rather conservative.
A last source of uncertainty is represented by the sampling design adopted for Val di Sole, with its particular topographic
and geological characteristics. The dominant southward aspect of the investigated rock glaciers, and their spatial
clustering, can explain the lack of correlation between water temperature and the aspect of rock glaciers. We tried to
minimise the spatial clustering of measured springs, visiting as many headwater catchments as possible, and taking
measurements at the largest number of springs on each catchment. However, due to logistic constraints and inherent
characteristics of the study area, a certain degree of spatial clustering was unavoidable. For this reason, the role of terrain
aspect as a possible controlling factor on spring-water temperature requires additional investigation.

**5.4 Geophysics**

The inverted resistivity sections obtained for the Preghena Rock Glacier (Fig. 8a and b) show results compatible with the
presence of permafrost patches. Even considering the high contact resistance due to the dry weather conditions preceding
the survey, and the location of the high resistivity body in the areas known to be the least sensitive of the model (the bed
and margins, Binley, 2015), we observe that the obtained resistivity values are typical of frozen materials (Hauck and
Kneisel, 2008). The high resistive area is highlighted by both ERT lines in the overlapping area (x<70 m in Line 1 and
x>100 m in Line2, Fig. 8). The data error of 20% applied in the inversion process was defined using the reciprocal
analysis, which minimise possible inversion artifacts compared to the more commonly used stacking error (Binley, 2015).
This result agrees with the low temperature of the Preghena Rock Glacier spring, which fluctuates between 1.6 and 1.8°C
throughout summer, and it suggests that this rock glacier should be classified as a pseudo-relict rock glacier.

In the Bordolona Rock Glacier (Fig. 8c), the frozen layer looks discontinuous in the lower section of the ERT Line, and more continuous and thicker in the upper part, where a younger lobe superposes the main body of the rock glacier. The different resistivity detected in the lower and upper sections of the ERT line can be related to a different percent ice content in the frozen layers, and/or a different temperature of the ice (Hilbich et al., 2008). These results suggest the probable presence of permafrost also inside the Bordolona Rock Glacier, which was considered a 'true' relict rock glacier due to its abundant vegetation cover, spring-water temperature above 3°C, and low mean elevation. Based on geophysical investigations, the Bordolona Rock Glacier too should be classified as a pseudo-relict rock glacier.

The acquired data were of lower quality at the Preghena Rock Glacier, due to the high contact resistance. More conclusive results should be obtained by repeating the geophysical surveys under wetter conditions, especially at the Preghena Rock Glacier, and possibly coupling ERT to seismic refraction measurements in order to obtain a reliable estimate of the percent ice content inside these rock glaciers (Hauck et al., 2008 and 2011, Wagner et al., 2019, Pavoni et al. 2023).

**5.5 Ice storage in the rock glaciers and glaciers of Val di Sole**

Calculations of the ice contained in the pseudo-relict rock glaciers of the study area assumed that 50% of the total area of relict rock glaciers contains permafrost (Section 4.2.1) and that the average ice content ranges between 5% and 20% in volume. This range is a first hypothesis based on the few geophysical data available at pseudo-relict rock glaciers (Delaloye, 2004; Colucci et al., 2019; Pavoni et al., 2023; this work). To our knowledge, the amount of ice in pseudo-relict rock glaciers has yet to be quantified.

Even if preliminary and affected by significant uncertainty, these estimates provide an order of magnitude of water stored as ice in the rock glaciers of Val di Sole. The water equivalent ratio for rock glacier ice versus glacier ice averages 1:4.1 and ranges between 1:3.6 and 1:4.8, considering minimum and maximum estimates reported above. Importantly, based on these calculations, 23% of the total rock glacier water volume would be stored inside pseudo-relict rock glaciers. Even assuming the lower bound of percent ice content (5%), pseudo-relict rock glaciers would contribute to a significant 9% of the total rock glacier water volume.

Based on the more conservative estimate reported in Section 5.2 for the frequency of pseudo-relict rock glaciers (38% instead of 50% of the total area covered by rock glaciers classified as relict), the water equivalent ratio for rock glacier ice versus glacier ice would average 1:4.3 and would range between 1:3.9 and 1:4.9, with 18% of the total rock glacier water volume stored inside pseudo-relict rock glaciers. Even if a little smaller, these numbers do not change significantly the meaning of the results.

The obtained water equivalent ratio of rock glacier ice versus glacier ice (between 1:4 and 1:5) is in the highest range of values reported in the literature for mountain regions where both glaciers and rock glaciers exist. Other studies in the European Alps (e.g., Barsch, 1977; Wagner et al., 2021) found ratios varying between 1:01 and 1:83, depending on catchment glacierization. A much larger range was reported for the Andes, comprised between 1:228 and 8.3:1. The largest ratios were found in arid regions of the Andean mountain range (Brenning, 2005; Azócar and Brenning, 2010; Rangecroft et al., 2015; Janke et al., 2017). Bolch and Marchenko (2009) reported ratios between 1:67 and 1:10 for the Northern Tien Shan, between Kazakhstan and Kyrgyzstan.

In Val di Sole, the ice volume of rock glaciers is already of the same order of magnitude of the ice contained in glaciers.
Considering that permafrost thaw rates are an order of magnitude or two slower compared to glacier ice (Hock et al.,
2019; Haeberli et al., 2017), and that about 3% of the glacier ice volume is depleted each year in the study area (Carturan
and De Blasi, 2021), the calculated ratio is expected to approach the unity within 2-3 decades.

## 6. Concluding remarks

We have surveyed spring-water temperature in an area of 795 km$^2$ in Val di Sole, to understand the influence of
topographic and geomorphological factors, and to test if it can be used to preliminary differentiate active/transitional and
relict rock glaciers. Spring-water temperature measurements enabled to characterise a large number of rock glaciers, and
to provide a first estimate of the frequency of pseudo-relict rock glaciers in this area. Overall, our results point to a
significant hydrological importance of rock glaciers classified as relict in the study area, which is expected to increase in
the future due to atmospheric warming.
In general, we have found that the spatial variability of spring-water temperature is controlled by elevation, aspect and
the presence of rock glaciers in the upslope area. Compared to other landforms in the upslope area, rock glaciers have
colder springs, irrespective of their activity state.
The spring-water temperature of rock glaciers classified so far as relict is higher and with larger spatial variability
compared to active/transitional rock glaciers. However, there is a remarkable ~50% (38% excluding extremes) overlap in
the spring temperature range of the two rock glacier groups. Relict rock glaciers tend to have colder springs if their surface
is blocky and scarcely covered by (cold-adapted) vegetation.
The spring-water temperature data suggest that one third of rock glaciers classified as relict might be actually pseudo-
relict, thus containing permafrost. The exact percentage cannot be derived unambiguously from spring-water temperature
because i) other evidence is required to confirm inference from water temperature, ii) there is uncertainty in the
classification of the activity state of rock glaciers, iii) there is geophysical evidence that rock glaciers containing
permafrost may have 'warm' springs (up to 3.7°C), and consequently iv) there is uncertainty in the definition of the
thresholds for differentiate among absent/possible/probable permafrost categories. We recommend further investigations
to reduce this uncertainty, for example performing geophysics on rock glaciers with a larger variability in surface
characteristics, activity, settings, and/or analysing the temporal variability of spring-water temperature.
Despite these uncertainties, our study shows that rock-glacier spring-water temperature can provide a pilot approach to
estimate the spatial distribution of permafrost in vast areas, and an auxiliary element to the classification of rock glaciers,
whose permafrost content might otherwise go underestimated. This method can be applied in other mountainous regions,
with the possible exception of arid/semi-arid regions where the presence of springs is scarce.
Geophysics applied to two rock glaciers classified as relict enabled to detect the presence of permafrost. While the blocky
Preghena Rock Glacier, whose spring temperature was < 1.8°C throughout the summer, was expected to contain
permafrost, its occurrence in the Bordolona Rock Glacier was not expected, because it is entirely covered by dense
vegetation and its spring temperature reached 3.7°C in late summer.
Preliminary calculations of water resources stored as ice inside the rock glaciers of Val di Sole reveal that they amount to
~24% of the water volume equivalent stored in glaciers, which are disappearing very fast. Remarkably, 20% of the total
rock glacier water volume is stored inside rock glaciers classified as relict.
This study highlights the need for additional investigations and improved understanding of these periglacial landforms.
In particular, the possible presence of permafrost in a large fraction of rock glaciers classified as relict poses critical
questions regarding the origin, preservation, current behaviour, seasonal dynamics, and future evolution of this
permafrost. Thorough study of pseudo-relict rock glaciers is required for understanding the evolution between active,
transitional and relict landforms, which is important in view of current and projected climate change.





**Appendix A**

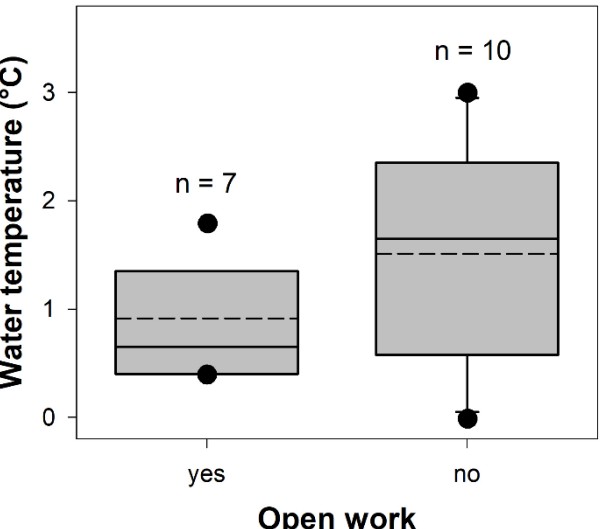


Figure A1: Spring-water temperature for intact rock glaciers with and without open work deposits on their surface. Boxes
indicate the 25th and 75th percentile, whiskers indicate the 10th and 90th percentile, whereas the horizontal solid and dashed
lines within the box mark the median and the mean, respectively. Maximum and minimum values are represented by dots.
Sample size (n) is reported above the boxplots.


**Data availability**

The spring-water temperature dataset used in this work is freely available by the Research Data Unipd repository (Carturan, 2024).

## Author contributions

LC designed the methodological approach and carried out the sampling campaigns with the support of AA, RS, MT, TZ and GZ. MP and JB carried out the geophysical surveys in cooperation with LC, CM and MZ and interpreted the results. GZ, LC and AA performed the statistical analyses of the dataset. LC prepared the first draft of the manuscript with contributions from GZ, MP and CM. All authors contributed to the editing of the manuscript.

## Competing interests

The contact author has declared that none of the authors has any competing interests.

## Acknowledgments

The authors acknowledge the editor and reviewers for their comments and suggestions.

## Financial support

This study was carried out within the RETURN Extended Partnership and received funding from the European Union Next-Generation EU (National Recovery and Resilience Plan – NRRP, Mission 4, Component 2, Investment 1.3 – D.D. 1243 2/8/2022, PE0000005), and the project PRIN 2022 "SUBSURFICE – Ecohydrological and environmental significance of subsurface ice in alpine catchments" (code: 2022AL7WKC, CUP: C53D23002020006), which received funding from the European Union Next-Generation EU (National Recovery and Resilience Plan – NRRP, Mission 4, Component 2, Investment 1.1 – D. D. 104 2/2/2022). This manuscript reflects only the authors' views and opinions; neither the European Union nor the European Commission can be considered responsible for them.

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
