# Peer review of "Spring-water temperature suggests widespread occurrence of"

_EGUsphere, 2023_

## Referee Comment (RC1)

**Spring-water temperature suggests widespread occurrence of Alpine permafrost in pseudo-relict rock glaciers**

Your manuscript presents the results from spring-water temperature of several rock glaciers, investigating in conjunction with topographic and geomorphological factors, spread in an area of 795 km$^2$. The underlying method is based on measuring spring-water temperature to distinguish between intact and relict rock glaciers. Only two specific cases are investigated with electrical resistivity tomography (ERT) to investigate the permafrost presence in the ground. Although the area investigated is commendable, the study presents methodological, conceptual, and formatting failures that make this work unsuitable for publication in its current shape.

GENERAL COMMENTS:

Background:

There is a lack of context with the previous work by Seppi et al. (2012), who pioneered the rock glaciers classification in the area investigated in this research work. In general, a detailed discussion between Seppi et al. (2012) and your contribution would allow placing your findings in the context of current research. Finally, there are very relevant limitations in this work in the points chosen for the spring-water temperature measurements which make some data used in the analyses not exactly reliable.

Methodology:

- The ERT surveys are performed only in two rock glaciers. Considering the aim of this work, this is not enough. There is too assumption about the spring-water temperature and the location of some of the measuring points which need to be verified not only on two rock glaciers. This is not enough to explain the difference in temperature in your dataset and cannot be used to discriminate intact rock glaciers from relict ones.
- There is a missing information about the runoff estimation. Did the runoff estimation do by visual inspection (as said in the line 171) or did you measure it properly?

Results:

- The subchapter "Ice storage in the rock glaciers and glaciers of Val di Sole" is placed in the wrong position. If the authors explain the methodology to estimate the ice volume and consequently the hydrological response, you should put these details in a proper subchapter in the methodology and add this notion in the introduction as well.

Discussion:

- Some information (see previous comment about the subchapter 5.5) are presented for the first time in this section. It is well explained, and the analysis are done in a proper way, but the position is wrong and completely unlinked with the text. The authors never mentioned previously this analysis, so its introduction is completely not in the correct place and never explained before along the text.

Limitations:

- As mentioned by the authors, there are several relevant limitations in this work.
  1. Location of the points where the spring-water temperature are performed.

If the point is located not in correspondence of the rock glacier but a few meters downstream, how this measure can be considered a real temperature value of the water coming out of the rock glacier? Between the rock glacier and the measuring point, the water is subject to alteration process that can alter its property and thus may represent an unrealistic spring-water temperature data. Therefore, this value should not be used to distinguish between intact and relict rock glaciers. By doing so, part of your dataset is based on unreliable data, if this situation arises. How many springs investigated fail in this case?

2. As you said, some springs were only monitored once. It seems a bit small to me to be used within a dataset where the ultimate goal is to use the spring-water temperature information to discriminate relict rock glaciers. I appreciate the explanation for their validation, but I do not think your conclusion to include this data in the dataset is robust enough.

3. This work seems based on outdate rock glacier classification which distinguish between intact (active and inactive), and relict rock glaciers. The update classification of rock glaciers distinguishes them between active, transitional, and relict. Could the authors explain why this latter classification is not taken into consideration?

DETAILS:

Line 60: check grammar

Line 86: ….and particularly on relict rock glaciers

Line 96: add reference

Line 107: what is the time interval considered for the precipitation parameter?

Lines 112-115: Add a short explanation of Seppi's method to classify rock glaciers and how they pointed out the number of rock glaciers in every single category.

Line 120: …mean annual precipitation of 1233 mm (Carturan et al., 2016)

Lines 129-130: rewrite the sentence. It is not clear.

Lines 130-132: These two sentences are not placed in a correct position since they provide results. Please, consider to move these lines in the appropriate section.

Line 133: What "considering accessibility" means?

Line 159: Why some springs were collected once per year? What is the reasoning behind the authors' choice to carry out only one measurement per year (albeit repeated between 2018 and 2020) and to consequently be able to consider this value sufficiently truthful?

Line 161: This classification between intact and relict rock glaciers is based on the outdate classification. As the authors may know, there is an update rock glaciers classification made by RGIK2023.

Line 171: How did the authors estimate runoff "visually"? It is a very subjective value and depends heavily on the operator in charge of the measurement.

Line 174: Do you mean outlier values?

Line 175: How did you exactly estimate runoff? You previously said, "runoff was visually estimated".

Line 177: See comment above. At least, insert a value for this "higher".

Line 219: I consider that one measure is not enough. Can the authors explain why they consider this measurement to be sufficient? Since your work is focused on distinguishing between intact and relict rock glaciers based on spring-water temperature, I don't think one measurement is sufficient enough.

Line 243: Insert the length of each survey.

Line 258: This seems more a result than a method. Consider moving in the appropriate section.

Lines 265-270: This part does not seem a result. It should be better to move it in the method section.

Line 386: Please consider indicating this information in Figure 1.

Lines 397-398: This is not something surprise. It has been already reported in some previously studies. Please add more recent references.

Section 5.5: Why this part is inserted in the discussion section? The explanation about the volumes should be moved in the methodological part.

Figure 2: Insert north arrow and scale bar.

Figure 3: Insert (a), (b), (c), and (d) and adjust the caption accordingly.

---

## Author Comment (AC1)

Dear Editor,

we would like to thank the two Reviewers for their careful reviews and for the suggestions that will help us improving considerably the manuscript.

In the following, we answer to the comments made by the reviewers, which have been numbered for improving clarity. The author responses are reported in blue colour right below the reviewers' comments. Line and page numbers are referred to the submitted paper.

**REVIEWER 1**

1.1) Your manuscript presents the results from spring-water temperature of several rock glaciers, investigating in conjunction with topographic and geomorphological factors, spread in an area of 795 km$^2$. The underlying method is based on measuring spring-water temperature to distinguish between intact and relict rock glaciers. Only two specific cases are investigated with electrical resistivity tomography (ERT) to investigate the permafrost presence in the ground. Although the area investigated is commendable, the study presents methodological, conceptual, and formatting failures that make this work unsuitable for publication in its current shape.

Based on the suggestions by Reviewer 1 we will revise the manuscript and hope it will be suitable for publication in the new shape.

1.2) There is a lack of context with the previous work by Seppi et al. (2012), who pioneered the rock glaciers classification in the area investigated in this research work. In general, a detailed discussion between Seppi et al. (2012) and your contribution would allow placing your findings in the context of current research.

In the Discussion section 5.2 we will add considerations regarding the Seppi et al. (2012) work and the need for a new (re)classification of rock glaciers based on the new data collected in our work. We thank the reviewer for pointing out this aspect, which was not included in the first version of the manuscript.

1.3) Finally, there are very relevant limitations in this work in the points chosen for the spring-water temperature measurements which make some data used in the analyses not exactly reliable.

Please refer to our replies to the 'Limitation' comments (points 1.9 to 1.11)

Methodology:

1.4) The ERT surveys are performed only in two rock glaciers. Considering the aim of this work, this is not enough.

Our paper is focused on spring-water temperature variability and uses geophysics as a mean for complementing spring-water temperature results at local scale for two rock glaciers. Geophysics is by no mean intended to characterise permafrost distribution in the study area, which would have been well beyond the objectives of this work. As reported in the Introduction, the general aim of the work is the analysis of the spatial variability of spring-water temperature in the study area to better understand permafrost distribution, under the hypothesis that a significant portion of rock glaciers classified as relict have spring-water temperature comparable to those of intact rock glaciers, as possible evidence of their pseudo-relict nature. The specific aim of geophysics is to investigate the presence of permafrost in two rock glaciers selected for their different spring-water temperature and surface characteristics, to constrain spring-water temperature results at local scale.

1.5) There is too assumption about the spring-water temperature and the location of some of the measuring points which need to be verified not only on two rock glaciers. This is not enough to explain the difference in temperature in your dataset and cannot be used to discriminate intact rock glaciers from relict ones.

We did not assume anything regarding spring-water temperature and the location of some of the measuring points. We measured spring-water temperature and analysed its spatial variability and the relationship with physical and morphometric variables. The measurement points were selected (where available and accessible) as representative of the rock glacier population in the study area (Section 3). We simply collected data and analysed them to check the hypothesis (not assumption) that part of rock glaciers classified as relict may have spring-water temperature comparable to intact rock glaciers.

Please refer to the reply to point 1.4 for considerations regarding geophysics. We do not understand how geophysics would allow verifying assumptions (if any) about spring-water temperature and the location of some measuring points. The relationship between spring-water temperature and permafrost is well known and has been long reported in the literature mentioned in this work.

1.6) There is a missing information about the runoff estimation. Did the runoff estimation do by visual inspection (as said in the line 171) or did you measure it properly?

The runoff was estimated visually and not measured. There was no need for accurate runoff measurements, because the aim of runoff estimation was only to discard springs with too low runoff. This approach is similar to that proposed e.g. by Strobl et al., (2020) for crowdsourced visual estimation of stream level class. We will add this in the text at the end of section 3.2.

Strobl, B., Etter, S., van Meerveld, I., & Seibert, J. (2020). Accuracy of crowdsourced streamflow and stream level class estimates. Hydrological Sciences Journal, 65(5), 823-841.

Results:

1.7) The subchapter "Ice storage in the rock glaciers and glaciers of Val di Sole" is placed in the wrong position. If the authors explain the methodology to estimate the ice volume and consequently the hydrological response, you should put these details in a proper subchapter in the methodology and add this notion in the introduction as well.

We will edit the manuscript so that this part will be placed in the proper sections (Introduction, Methods, Results and Discussion).

Discussion:

1.8) Some information (see previous comment about the subchapter 5.5) are presented for the first time in this section. It is well explained, and the analysis are done in a proper way, but the position is wrong and completely unlinked with the text. The authors never mentioned previously this analysis, so its introduction is completely not in the correct place and never explained before along the text.

Please see the reply to the previous comment 1.7

Limitations:

1.9) First limitation: as mentioned by the authors, there are several relevant limitations in this work.

Location of the points where the spring-water temperature are performed.

If the point is located not in correspondence of the rock glacier but a few meters downstream, how this measure can be considered a real temperature value of the water coming out of the rock glacier? Between the rock glacier and the measuring point, the water is subject to alteration process that can alter its property and thus may represent an unrealistic spring-water temperature data. Therefore, this value should not be

used to distinguish between intact and relict rock glaciers. By doing so, part of your dataset is based on unreliable data, if this situation arises. How many springs investigated fail in this case?

We have checked the impact of spring location downstream of the rock glacier front at three measurement sites, where the same stream emerged briefly at the rock glacier front and a few tens meters downstream. Measurements confirmed that there is negligible warming (from 0.0 to 0.1°C) of the water downstream of the rock glacier front, at least as long as the water remains below the surface (this will be added in the text in Section 5.3). Possible warming over longer distance is what happens at the Bordolona rock glacier, as discussed in section 5.3. However, given the patchy distribution of permafrost in pseudo-relict rock glaciers, there is no way to know or estimate the distance between permafrost and the measured springs, as this would require systematic and extensive geophysical surveys on all rock glaciers where spring-water temperature is measured.

1.10) Second limitation: as you said, some springs were only monitored once. It seems a bit small to me to be used within a dataset where the ultimate goal is to use the spring-water temperature information to discriminate relict rock glaciers. I appreciate the explanation for their validation, but I do not think your conclusion to include this data in the dataset is robust enough.

We thank the reviewer for her/his opinion but we prefer to quantify uncertainty by showing experimental data and numeric validation in our manuscript. Please see also the reply to comment 1.27.

1.11) Third limitation: this work seems based on outdate rock glacier classification which distinguish between intact (active and inactive), and relict rock glaciers. The update classification of rock glaciers distinguishes them between active, transitional, and relict. Could the authors explain why this latter classification is not taken into consideration?

As we explain in the text, the only available classification was that of Seppi et al. (2012), who inventoried the rock glaciers in the study area using guidelines available in 2012. We will edit the text and the figures in order to account for the updated classification proposed by RGIK (2023).

RGIK (2023). Guidelines for inventorying rock glaciers: baseline and practical concepts (version 1.0). IPA Action Group Rock glacier inventories and kinematics, 25 pp, DOI: 10.51363/unifr.srr.2023.002

DETAILS:

1.12) Line 60: check grammar

We will divide this period into two separate sentences

1.13) Line 86: ….and particularly on relict rock glaciers

We prefer adding 'of', instead of 'on', because it better clarifies the meaning of this sentence

1.14) Line 96: add reference

We will add the requested references (geological map of Italy)

1.15) Line 107: what is the time interval considered for the precipitation parameter?

The time interval goes from 1971 to 2008, added in the text

1.16) Lines 112-115: Add a short explanation of Seppi's method to classify rock glaciers and how they pointed out the number of rock glaciers in every single category.

Ok, it will be added (end of Section 2)

1.17) Line 120: …mean annual precipitation of 1233 mm (Carturan et al., 2016)

Modified accordingly (beginning of Section 3.1)

1.18) Lines 129-130: rewrite the sentence. It is not clear.

We will simplify this sentence for clarity

1.19) Lines 130-132: These two sentences are not placed in a correct position since they provide results. Please, consider to move these lines in the appropriate section.

Actually these sentences help the reader understanding what follows, that is why we built the spring-water sampling scheme around rock glacier activity, length, mean elevation, and vegetation cover.

1.20) Line 133: What "considering accessibility" means?

Accessibility 'of springs', it will be added in the text to improve clarity

1.21) Line 159: Why some springs were collected once per year? What is the reasoning behind the authors' choice to carry out only one measurement per year (albeit repeated between 2018 and 2020) and to consequently be able to consider this value sufficiently truthful?

Most springs were measured once per year between 2018 and 2020, only a few of them was also measured in 2021. Only one measurement per year was feasible at each spring, primarily because of the large number of springs measured over a large area, short period of time available (about one month), limited funding and, in particular, low available manpower (often only one person). We think that there is no need to write it out explicitly.

1.22) Line 161: This classification between intact and relict rock glaciers is based on the outdate classification. As the authors may know, there is an update rock glaciers classification made by RGIK2023.

Please see the reply to the comment 1-11 (Third limitation)

1.23) Line 171: How did the authors estimate runoff "visually"? It is a very subjective value and depends heavily on the operator in charge of the measurement.

Yes it is but the operator was always the same. We add that runoff estimation was only used to discard measurements at springs with too low runoff (please, see also the reply to comment 1.6).

1.24) Line 174: Do you mean outlier values?

No, we mean problems of too low runoff and selection of only one measurement for each rock glacier, where there was redundancy (as detailed in the following)

1.25) Line 175: How did you exactly estimate runoff? You previously said, "runoff was visually estimated".

There was no need for exact runoff estimation, it should be clearer with the explaining text added at the end of Section 3.2

1.26) Line 177: See comment above. At least, insert a value for this "higher".

Here 'higher' is a relative concept: higher compared to the other springs available at the same rock glacier. There is no need for inserting runoff values in our opinion.

1.27) Line 219: I consider that one measure is not enough. Can the authors explain why they consider this measurement to be sufficient? Since your work is focused on distinguishing between intact and relict rock glaciers based on spring-water temperature, I don't think one measurement is sufficient enough.

We do not write that 'one measure is enough'. Conversely, we express doubt about that and question explicitly the representativeness of springs measured only once. To prove/disprove that, we present

calculation results based on the collected data, that indicate a low impact of extreme temperatures and the suitability of using the median of all available measurements (regardless of their number) in statistical analyses. We let the numbers speak for themselves because our personal opinion is not relevant in a scientific paper.

Most of the papers regarding permafrost applications using spring-water temperature mentioned in our manuscript deal with single measurements, including our published paper based on this methodology (Carturan et al., 2016). For this reason, we consider a plus that the majority of springs analysed in Val di Sole had more than one measurement! Actually, we planned to repeat all the measurements at least three times, in three different years, but many springs had too low runoff or were even dry at the time of revisits, due to drought conditions.

1.28) Line 243: Insert the length of each survey.

Ok, it will be added in Section 3.4

1.29) Line 258: This seems more a result than a method. Consider moving in the appropriate section.

Ok, this part will be moved at the end of Section 5.4

1.30) Lines 265-270: This part does not seem a result. It should be better to move it in the method section.

Here we shortly present the descriptive statistics of our dataset reported in Table 4. For this reason, we prefer keeping this part where it is.

1.31) Line 386: Please consider indicating this information in Figure 1.

Ok, we will add the location of Careser diga in Figure 1

1.32) Lines 397-398: This is not something surprise. It has been already reported in some previously studies. Please add more recent references.

Ok, we will add these references:

Wagner, T., Pauritsch, M., Mayaud, C., Kellerer-Pirklbauer, A., Thalheim, F., & Winkler, G. (2019). Controlling factors of microclimate in blocky surface layers of two nearby relict rock glaciers (Niedere Tauern Range, Austria). Geografiska Annaler: Series A, Physical Geography, 101(4), 310–333. https://doi.org/10.1080/04353676.2019.1670950

Amschwand, D., Scherler, M., Hoelzle, M., Krummenacher, B., Haberkorn, A., Kienholz, C. and Gubler, H., 2024. Surface heat fluxes at coarse blocky Murtèl rock glacier (Engadine, eastern Swiss Alps). The Cryosphere, 18(4), pp.2103-2139.

1.33) Section 5.5: Why this part is inserted in the discussion section? The explanation about the volumes should be moved in the methodological part.

Ok, we will modify this accordingly

1.34) Figure 2: Insert north arrow and scale bar.

Ok, we will add north arrow and scale bar in Figure 2

1.35) Figure 3: Insert (a), (b), (c), and (d) and adjust the caption accordingly.

Ok, we will modify this accordingly

---

## Author Comment (AC2)

Dear Editor,

we would like to thank the two Reviewers for their careful reviews and for the suggestions that will help us improving considerably the manuscript.

In the following, we answer to the comments made by the reviewers, which have been numbered for improving clarity. The author responses are reported in blue colour right below the reviewers' comments. Line and page numbers are referred to the submitted paper.

**REVIEWER 2**

RC2: 'Comment on egusphere-2023-2689', Cristian Daniel Villarroel, 28 Jun 2024 reply

Dear Authors,

2.1) I have read in detail the work entitled ¨Spring-water temperature suggests widespread occurrence of Alpine permafrost in pseudo-relict rock glaciers ¨. Personally, I consider that it is a work of very good quality and according to the readers of this journal. The research topic is of particular interest in the Alps, and also in other mountainous regions. The use of spring-water temperature is a method that can provide valuable information on the presence and distribution of mountain permafrost, which is an issue where there are current uncertainties, mainly in the relict/pseudo-relict category. One of the advantages of this method is that it is relatively easy (the measurement device is light) to perform on the ground, regardless of the climatic and topographic difficulties of all mountainous regions. In addition, registered values, which are not conclusive by themselves, can help in the decision of other methods to apply and in the selection of the sites to be measured. This method can be applied in other mountainous regions, but I consider that in arid/semi-arid regions the presence of springs is scarce to cover spatial variability.

This is an important point raised by the Reviewer, we will add this latest consideration in the manuscript (Discussion Section)

2.2) On the other hand, I consider that the spring-water temperature analysis is a complementary method in studying the presence and distribution of mountain permafrost. The application only of this method would generate ambiguous results. Warm temperatures (such as some registered in this work) are not enough to rule out the presence of permafrost. For their part, cold temperatures, they would not be enough to confirm the presence of permafrost. In this last point it is important to highlight that the ground ice stored seasonally in active layer can have influence on the temperature of the springs. Therefore, it is highly recommended to carry out other methods. EC and TDS measurement and hydrochemical and isotopic analysis could provide valuable information.

We agree with the reviewer about the importance of integrating different methods to verify the role of permafrost on the hydrology of springs in mountain areas. However, based on our data collected in a sub-catchment of Val di Sole, only spring-water temperature proved to be effective in discriminating between permafrost affected and non-permafrost affected springs, whereas EC and isotopes where not useful (Carturan et al., 2016).

We agree with the reviewer that warm temperatures and cold temperatures alone are not sufficient to infer respectively the absence and presence of permafrost. Warm temperatures are already discussed this in

Section 5. Regarding cold temperatures due to possible seasonal ice in the ground, please see the reply to the following comment 2.6.

2.3) ERT profiles made in this work contribute to determining the presence or not of ground ice in the two rock glaciers studied. However, in one of them (Preghena) the resistive anomaly that would indicate the presence of ground ice is located at the 2D profile edges. In these sectors the reliability of the results decreases considerably. In addition, in this same rock glacier, a 20% error is considerably high.

We agree with the comment of the reviewer that the bottom and the edges of the tomogram are the least sensitive zones, nevertheless we acquired a large number of measurements with the dipole-dipole multi-skip scheme (about 5200 quadrupoles) and, even after the applied filtering, the pseudo-section was homogeneously covered by apparent resistivities, including the edges and the bottom. We are confident we are not assessing artifacts, because the high resistive area is highlighted by both the ERT lines in the overlapping position (x<70 m in Line 1 and x>100 m in Line2). We must specify better that the data error of 20% applied in the inversion process was defined using the reciprocal analysis, which minimize possible inversion artifacts (Binley, 2015), and not the more common stacking error. Obviously, the expected data error can be estimated also with the stacking error (we acquired the measurements with a stacking range between 3-6, and a standard deviation threshold of 5%), nevertheless this approach usually overestimates the quality of the dataset and is less reliable (Binley, 2015).

We will add these considerations in Section 5.4.

2.4) In general, the work is well structured, the reading is pleasant, the data is relevant, and the figures are of good quality and enriching.

Specific comments are made below.

Line 13: This work is not focused on the water contribution of the talus slopes. Nor are there many background to mention the importance or not of the talus slopes in the water supply to the rivers. Where does this statement arise?

Here we will remove the mention to talus slopes and rephrase slightly this sentence.

2.5) Line 68-71: InSAR or DInSAR would not be useful techniques for this case since the relict or pseudo relict rock glaciers have no movement or the movement can be very slow and be in the same range as the uncertainty of the method. For this reason, since it is not a method applicable to this study, this sentence should be removed.

We will add these consideration in the Introduction, where we write about InSAR. We would prefer keeping this sentence because some reader may wonder if we have considered this methodology.

2.6) Line 73-75: The measurement could be at the end of summer or principles of autumn. The temperature trend in every summer should be considered, because with colder summers the thermal wave takes longer to enter the subsoil and completely defrost the active layer. This research methodology assumes that the water in which the temperature is being measured is influenced by ground ice. But, so that the temperature value registered in the springs is a real evidence of permafrost existence, it should not exist seasonal ice in the active layer.

The reviewer is right that cold temperatures can be due to the presence of seasonal ground ice. For this reason, we have stuck with the standard procedure of performing measurements at the end of the summer/beginning of autumn in order to exclude seasonal ground-ice influence on temperature measurements, or at least minimise its effect within the limit of our approach. It is true that seasonal ground ice formation could have a stronger influence on spring temperature after cool/short summer seasons, but

this was not the case in the years we have analysed, characterised by warm/long summers. We will add these considerations in the Discussion section (paragraph 5.3).

2.7) Line 94-96: It should be specified that lithology is composed of each rock glacier. Considering that ERT has been applied, which are of the relict or pseudo-relict type, and that the amount of ice present can be small, the changes or gradients in the values of the electrical resistivity could be influenced by lithology. This information could also be added in section 3.4.

The inspected rock glaciers are characterised by uniform lithology. Information regarding lithology will be added in the text.

2.8) Line 156 (Data collection): Has a monitoring of temperature variation in springs during the day been performed? If there is a variation of temperature during the day this will influence the results depending on the measurement time.

Hourly spring water records collected during summer and early autumn using dataloggers show negligible variation of spring-water temperature during the day (Seppi, 2006). We will add these considerations in Section 5.3.

Seppi, R.: I rock glaciers delle Alpi Centrali come indicatori ambientali (Gruppo Adamello-Presanella e settore orientale del Gruppo Ortles-Cevedale) - Rock glaciers of the Central Alps as environmental indicators (Adamello-Presanella Group and eastern sector of the Ortles-Cevedale Group). Phd Thesis, 199 pp., doi: 10.13140/RG.2.1.1186.5682, 2006.

2.9) Line 161-162: It is not completely clear to me. In the previous paragraph it is mentioned that 133 springs located downslope of rock glaciers were measured. This paragraph says that 67 (17 + 50) springs in rock glaciers were measured. What is the amount of spings in rock glaciers that have been measured?

Here we report how many rock glaciers we sampled, not how many springs downslope of rock glaciers. Before, we mention 133 springs downslope of rock glaciers because there were rock glaciers with multiple springs measured. We will add this at the beginning of Section 3.2, to improve clarity. Afterwards, in Section 3.3, we clarify that we retained only one spring for each rock glacier sampled.

2.10) Line 175: How were those streamflows measured? With what criteria was that threshold (0.11 l/s) established?

We have a long experience of runoff measurements using the salt dilution method on mountain creeks and torrents. Based on this experience we were able to estimate by eye the runoff of springs, and to assess that a 0.1 l/s threshold was adequate for discriminating between semi-stagnant and well-fed springs, and thus to discard springs affected by large temperature fluctuations during the day. The estimated runoff was not used in the analyses of spring-water temperature. This will be added in the text at the beginning of Section 3.3. Please see also the reply to comment 1.6.

2.11) Line 251-252: In these cases of high contact resistance it is advisable to add under the sponge an aluminum foil of approximately 20 cm side and cover it if possible with fine material. If contact resistances remain high, after incorporating abundant salt water and aluminum foil, it is advisable to move the profile position to a more favorable sector. Personally, I do not recommend measuring with such high contact resistances.

We agree with the Reviewer that these were unfavourable conditions, even applying the traditional approach of stainless-steel spike coupled with sponges soaked in salt water (Hauck and Kneisel, 2008), which usually guarantee optimal contact resistances (<100 kΩ). Nevertheless, in the specific case of the Preghena rock glacier, the surface was particularly dry due to drought conditions. In our experience, good quality ERT datasets can be collected on Alpine rock glaciers even with contact resistances larger than 100 kΩ, adopting large quadrupoles number acquisitions that considerably helps redundancy and data reliability, as we did.

The surface of the rock glacier is almost entirely covered by blocks. For this reason, it was not possible to find a more favourable position.

2.12) Figure 8: The scale of values should be expressed in Ωm or kΩm, as in the text. In addition, it is convenient to add the RMS in each profile.

We will make uniform the units of measurement between the text and the figure. We will also add the RMS in each profile, as suggested.

2.13) Line 464 (section 5.3): One of the main uncertainties is that it is not known if there is ground ice from the previous winter that has been stored in an active layer and that is influencing the temperature of the springs. In a way, the monitoring of the springs over the years would allow to eliminate this situation to a certain degree.

Please see the reply to the previous comment 2.6. We will add these relevant considerations in Section 5.3.

2.14) Another uncertainty factor is that water inside a rock glacier can follow different paths (Villarroel et al., 2022). In pseudo-relict rock glaciers, with the presence of ground ice in the form of islands, the water could follow paths without contact with the ice. I consider that this situations must be discussed.

We agree with the Reviewer that this situation must be included in the discussion, and we will do that in the revised version of our manuscript.

---

## Author Response (AR1)

Dear Editor,

We thank the Reviewers for their comments and insights. We have now changed the manuscript following their suggestions, which allowed us to clarify several points in the paper. Major modifications applied were:

- use of the updated rock glacier classification according to the RGIK (2023) guidelines throughout the manuscript (text, tables, figures)
- better clarification of the specific aims in the introduction
- the former section 5.5 "Ice storage in the rock glaciers and glaciers of Val di Sole" was split among methods (new Section 3.5), results (new Section 4.4) and discussions (Section 5.5)
- additional limitations of our methodological approach have been added in Section 5.3
- Figures 1, 2, 3, 6, 8, 9 have been redone according to the reviewers' suggestions
- various edits as detailed in the replies to the reviewers' specific comments

The dataset has been uploaded in an open-access repository, currently under validation by an editor of the researchdata.cab.unipd.it repository (DOI: 10.25430/researchdata.cab.unipd.it.00001366)

In the following, we report a point-by-point reply to the comments made by the reviewers, which have been numbered for improving clarity. Our responses are reported in green colour right below the reviewers' comments. Line and page numbers in the reviewers' comments are referred to the submitted paper, whereas line numbers in our (green) replies are referred to the 'clean' version of the revised manuscript.

**REVIEWER 1**

1.1) Your manuscript presents the results from spring-water temperature of several rock glaciers, investigating in conjunction with topographic and geomorphological factors, spread in an area of 795 km$^2$. The underlying method is based on measuring spring-water temperature to distinguish between intact and relict rock glaciers. Only two specific cases are investigated with electrical resistivity tomography (ERT) to investigate the permafrost presence in the ground. Although the area investigated is commendable, the study presents methodological, conceptual, and formatting failures that make this work unsuitable for publication in its current shape.

1.2) There is a lack of context with the previous work by Seppi et al. (2012), who pioneered the rock glaciers classification in the area investigated in this research work. In general, a detailed discussion between Seppi et al. (2012) and your contribution would allow placing your findings in the context of current research.

In the Discussion section 5.2 (L. 476-479) we added considerations regarding the Seppi et al. (2012) work and the need for a new (re)classification of rock glaciers based on the new data collected in our work. We thank the reviewer for pointing out this aspect, which was not included in the first version of the manuscript.

1.3) Finally, there are very relevant limitations in this work in the points chosen for the spring-water temperature measurements which make some data used in the analyses not exactly reliable.

Please refer to our replies to the 'Limitation' comments (points 1.9 to 1.11)

Methodology:

1.4) The ERT surveys are performed only in two rock glaciers. Considering the aim of this work, this is not enough.

Our work is focused on spring-water temperature variability and uses geophysics as a mean for complementing spring-water temperature results at local scale for two rock glaciers. Geophysics is by no mean intended to characterise permafrost distribution in the study area, which would have been well beyond

the objectives of this work. As reported at the end of the Introduction (L. 88-94), the general aim of the work is the analysis of the spatial variability of spring-water temperature in the study area to better understand permafrost distribution, hypothesising that a significant portion of rock glaciers classified as relict have spring-water temperature comparable to those of intact rock glaciers, as possible evidence of their pseudo-relict nature. The specific aim of geophysics is to investigate the presence of permafrost in two rock glaciers classified as relict and selected for their different spring-water temperature and surface characteristics, to constrain spring-water temperature results at local scale. This was remarked for clarity at the beginning of Section 3.4 in the Methods.

1.5) There is too assumption about the spring-water temperature and the location of some of the measuring points which need to be verified not only on two rock glaciers. This is not enough to explain the difference in temperature in your dataset and cannot be used to discriminate intact rock glaciers from relict ones.

In our opinion, we actually avoided assumptions regarding spring-water temperature and the location of some of the measuring points. We measured spring-water temperature and analysed its spatial variability and the relationship with physical and morphometric variables. The measurement points were selected (where available and accessible) as representative of the rock glacier population in the study area (Section 3). The collected data were analysed to check the hypothesis that part of rock glaciers classified as relict might have spring-water temperature comparable to intact rock glaciers, as a possible evidence of their permafrost content. The relationship between spring-water temperature and permafrost is well known and has been long reported in the literature mentioned in our manuscript (e.g., Haeberli, 1975; Frauenfelder et al., 1998; Scapozza, 2009). For these reasons, we think that this approach is evidence-based and intended to minimise assumptions. Please refer to the reply to point 1.4 for considerations regarding geophysics.

1.6) There is a missing information about the runoff estimation. Did the runoff estimation do by visual inspection (as said in the line 171) or did you measure it properly?

The runoff was estimated visually and not measured. There was no need for accurate runoff measurements, because the aim of runoff estimation was only to discard springs with too low runoff. We added this in the text at the end of section 3.2 (L 180-182). This approach is similar to that proposed e.g. by Strobl et al., (2020) for crowdsourced visual estimation of stream level class.

Strobl, B., Etter, S., van Meerveld, I., & Seibert, J. (2020). Accuracy of crowdsourced streamflow and stream level class estimates. Hydrological Sciences Journal, 65(5), 823-841.

Results:

1.7) The subchapter "Ice storage in the rock glaciers and glaciers of Val di Sole" is placed in the wrong position. If the authors explain the methodology to estimate the ice volume and consequently the hydrological response, you should put these details in a proper subchapter in the methodology and add this notion in the introduction as well.

Agreed. The manuscript has been edited and section 5.5 has been split accordingly to Reviewer 1's request. This concept and its importance are already mentioned at the beginning of the introduction (L. 35-44), we have added a specific aim at the end of the introduction section (L. 94) regarding ice content preliminary estimations. The remaining part is now divided between methods (new Section 3.5), results (new Section 4.4) and discussions (Section 5.5), with minor changes and adjustments. In particular, in Section 5.5 we have added a comparison with two different hypotheses regarding the frequency of pseudo-relict rock glaciers (38% and 50% of all rock glaciers classified as relict) based on what is reported in the results (Section 4.2.1) and in the discussion (Section 5.2).

Discussion:

1.8) Some information (see previous comment about the subchapter 5.5) are presented for the first time in this section. It is well explained, and the analysis are done in a proper way, but the position is wrong and completely unlinked with the text. The authors never mentioned previously this analysis, so its introduction is completely not in the correct place and never explained before along the text.

Please see the reply to the previous comment 1.7

Limitations:

1.9) First limitation: as mentioned by the authors, there are several relevant limitations in this work.

Location of the points where the spring-water temperature are performed.

If the point is located not in correspondence of the rock glacier but a few meters downstream, how this measure can be considered a real temperature value of the water coming out of the rock glacier? Between the rock glacier and the measuring point, the water is subject to alteration process that can alter its property and thus may represent an unrealistic spring-water temperature data. Therefore, this value should not be used to distinguish between intact and relict rock glaciers. By doing so, part of your dataset is based on unreliable data, if this situation arises. How many springs investigated fail in this case?

We agree with the Reviewer that this source of uncertainty was not adequately discussed in the first version of the manuscript. We have checked the impact of spring location downstream of the rock glacier front at three measurement sites, where the same stream emerged briefly at the rock glacier front and a few tens meters downstream. Measurements confirmed that there is negligible warming (from 0.0 to 0.1°C) of the water downstream of the rock glacier front, at least as long as the water remains below the surface. This evidence has now been added in the text in Section 5.3 (L. 539-542).

Warming over longer distance at the Bordolona rock glacier was already discussed in section 5.3 (L. 532-537). This has been further clarified specifying which distance we refer to (more than one hundred meters) and adding the case of patchy permafrost, not in contact with groundwater paths.

1.10) Second limitation: as you said, some springs were only monitored once. It seems a bit small to me to be used within a dataset where the ultimate goal is to use the spring-water temperature information to discriminate relict rock glaciers. I appreciate the explanation for their validation, but I do not think your conclusion to include this data in the dataset is robust enough.

We thank the reviewer for her/his feedback regarding this relevant point, which is addressed at the end of Section 3.3 (L. 232-237), quantifying uncertainties and impacts of single (and possibly extreme) measurements. Please see the reply to comment 1.27 for further considerations.

1.11) Third limitation: this work seems based on outdate rock glacier classification which distinguish between intact (active and inactive), and relict rock glaciers. The update classification of rock glaciers distinguishes them between active, transitional, and relict. Could the authors explain why this latter classification is not taken into consideration?

In the previous version of the manuscript, we based our classification referring to the only available classification was that of Seppi et al. (2012), who inventoried the rock glaciers in the study area using guidelines available in 2012. We have now edited the text and the figures/tables replacing the adjective 'intact' with 'active/transitional' in order to account for the updated classification proposed by RGIK (2023).

RGIK (2023). Guidelines for inventorying rock glaciers: baseline and practical concepts (version 1.0). IPA Action Group Rock glacier inventories and kinematics, 25 pp, DOI: 10.51363/unifr.srr.2023.002

DETAILS:

1.12) Line 60: check grammar

Agreed. We divided this period into two separate sentences

1.13) Line 86: ....and particularly on relict rock glaciers

We prefer adding 'of', instead of 'on', because it better clarifies the meaning of this sentence

1.14) Line 96: add reference

Agreed. We added the requested references (geological map of Italy: Dal Piaz et al., 2007; Martin et al., 2009; Chiesa et al., 2010, Montrasio et al., 2012)

1.15) Line 107: what is the time interval considered for the precipitation parameter?

The time interval goes from 1971 to 2008, added in the text (L. 114).

1.16) Lines 112-115: Add a short explanation of Seppi's method to classify rock glaciers and how they pointed out the number of rock glaciers in every single category.

Ok, added (end of Section 2, L. 118-122)

1.17) Line 120: ...mean annual precipitation of 1233 mm (Carturan et al., 2016)

Agreed. Modified accordingly (L. 127)

1.18) Lines 129-130: rewrite the sentence. It is not clear.

Agreed. We simplified this sentence for clarity: "The aim was to evaluate their possible covariance and to optimise the number of variables and their combinations, to be included in the sampling scheme"

1.19) Lines 130-132: These two sentences are not placed in a correct position since they provide results. Please, consider to move these lines in the appropriate section.

We would prefer to keep the sentences in this section as they clarify the reasons why we built the spring-water sampling scheme around rock glacier activity, length, mean elevation, and vegetation cover.

1.20) Line 133: What "considering accessibility" means?

Accessibility 'of springs', added in the text to improve clarity (L. 140)

1.21) Line 159: Why some springs were collected once per year? What is the reasoning behind the authors' choice to carry out only one measurement per year (albeit repeated between 2018 and 2020) and to consequently be able to consider this value sufficiently truthful?

Most springs were measured once per year between 2018 and 2020, only a few of them were also measured in 2021. Only one measurement per year was feasible at each spring, primarily because of the large number of springs measured over a large area, short period of time available (about one month), limited funding and, in particular, low available manpower (often only one person). We think that there is no need to write it out explicitly.

Papers dealing with rock glacier spring-water temperature measurements (e.g. Frauenfelder et al., 1998; Imhof et al., 2000; Strozzi et al., 2004; Cossart et al., 2008, mentioned in the Introduction) generally report 'late-summer' measurements, without explicitating measurement dates and number of annual measurements for each spring. Scapozza (2009) in Table 3 reports single measurements at each spring for each year (as we did). Haeberli (1985) reports that "(rock glacier) temperature does not seem to change markedly, be it in seasonal cycles or even in time periods of deacades (Haeberly 1975; Haeberli and Patzelt

1983).” Our former paper on this argument (Carturan et al., 2016) was based on single yearly measurements of spring water temperature for each site.

Haeberli, W.: Creep of mountain permafrost: internal structure and flow of Alpine rock glaciers, Mitteilung VAW/ETHZ, 77, 142 pp., 1985.

1.22) Line 161: This classification between intact and relict rock glaciers is based on the outdate classification. As the authors may know, there is an update rock glaciers classification made by RGIK2023.

Please see the reply to the comment 1.11 (Third limitation)

1.23) Line 171: How did the authors estimate runoff “visually”? It is a very subjective value and depends heavily on the operator in charge of the measurement.

We agree with the reviewer that this can be subjective, but the operator was always the same. We modified this sentence as follows (L. 180-182): “In addition, we assessed runoff by a quick visual estimation (always the same operator) similar to Strobl et al. (2020), who considered average width, mean depth and velocity of the flow downslope of the spring. This approach was used to rule out springs with very low runoff (<0.1 l/s)”. Please, see also the reply to comment 1.6.

1.24) Line 174: Do you mean outlier values?

We mean problems of too low runoff and selection of only one measurement for each rock glacier, required where there was redundancy (as detailed in the following sentence of the manuscript)

1.25) Line 175: How did you exactly estimate runoff? You previously said, “runoff was visually estimated”.

There was no need for exact runoff estimation, it should be clearer now with the explaining text added in the last sentence of Section 3.2 (L. 180-182).

1.26) Line 177: See comment above. At least, insert a value for this “higher”.

Here ‘higher’ is a relative concept: higher compared to the other springs available at the same rock glacier. We have replaced with ‘highest’, ‘closest’, ‘lowest’, which clarify what we mean (L. 189-190)

1.27) Line 219: I consider that one measure is not enough. Can the authors explain why they consider this measurement to be sufficient? Since your work is focused on distinguishing between intact and relict rock glaciers based on spring-water temperature, I don't think one measurement is sufficient enough.

In the manuscript (end of Section 3.3, L. 233-238) we express our doubts about the limitation of one measurement and explicitly question the representativeness of springs measured only once. To prove/disprove that, we present calculation results based on the collected data, that indicate a low impact of extreme temperatures and the suitability of using the median of all available measurements (regardless of their number) in statistical analyses.

We planned to repeat all the measurements at least three times, in three different years, but several springs had too low runoff or were even dry at the time of revisits, due to drought conditions. We consider a plus that the majority of springs analysed in Val di Sole had more than one measurement.

1.28) Line 243: Insert the length of each survey.

Agreed. Added in Section 3.4 (L. 257-258)

1.29) Line 258: This seems more a result than a method. Consider moving in the appropriate section.

We agree and moved this part in the last period of Section 5.4 (L. 589).

1.30) Lines 265-270: This part does not seem a result. It should be better to move it in the method section.

Here we shortly present the descriptive statistics of our dataset reported in Table 4. For this reason, we prefer keeping this part where it is.

1.31) Line 386: Please consider indicating this information in Figure 1.

Agreed. We have added the location of Careser diga in Figure 1

1.32) Lines 397-398: This is not something surprise. It has been already reported in some previously studies. Please add more recent references.

Agreed, we have now added these references (Section 5.1, L. 441)):

Wagner, T., Pauritsch, M., Mayaud, C., Kellerer-Pirklbauer, A., Thalheim, F., & Winkler, G. (2019). Controlling factors of microclimate in blocky surface layers of two nearby relict rock glaciers (Niedere Tauern Range, Austria). Geografiska Annaler: Series A, Physical Geography, 101(4), 310–333. https://doi.org/10.1080/04353676.2019.1670950

Amschwand, D., Scherler, M., Hoelzle, M., Krummenacher, B., Haberkorn, A., Kienholz, C. and Gubler, H., 2024. Surface heat fluxes at coarse blocky Murtèl rock glacier (Engadine, eastern Swiss Alps). The Cryosphere, 18(4), pp.2103-2139.

1.33) Section 5.5: Why this part is inserted in the discussion section? The explanation about the volumes should be moved in the methodological part.

Ok, modified accordingly (see the reply to point 1.7)

1.34) Figure 2: Insert north arrow and scale bar.

We have added the north arrow and scale bar in Figure 2, as suggested

1.35) Figure 3: Insert (a), (b), (c), and (d) and adjust the caption accordingly.

We have modified this figure and caption accordingly

**REVIEWER 2**

RC2: 'Comment on egusphere-2023-2689', Cristian Daniel Villarroel, 28 Jun 2024 reply

Dear Authors,

2.1) I have read in detail the work entitled ¨Spring-water temperature suggests widespread occurrence of Alpine permafrost in pseudo-relict rock glaciers ¨. Personally, I consider that it is a work of very good quality and according to the readers of this journal. The research topic is of particular interest in the Alps, and also in other mountainous regions. The use of spring-water temperature is a method that can provide valuable information on the presence and distribution of mountain permafrost, which is an issue where there are current uncertainties, mainly in the relict/pseudo-relict category. One of the advantages of this method is that it is relatively easy (the measurement device is light) to perform on the ground, regardless of the climatic and topographic difficulties of all mountainous regions. In addition, registered values, which are not conclusive by themselves, can help in the decision of other methods to apply and in the selection of the sites to be measured. This method can be applied in other mountainous regions, but I consider that in arid/semi-arid regions the presence of springs is scarce to cover spatial variability.

This is an important point raised by the Reviewer, we added this latest consideration in the manuscript (Section 6, Concluding remarks, L. 635-636)

2.2) On the other hand, I consider that the spring-water temperature analysis is a complementary method in studying the presence and distribution of mountain permafrost. The application only of this method would generate ambiguous results. Warm temperatures (such as some registered in this work) are not enough to rule out the presence of permafrost. For their part, cold temperatures, they would not be enough to confirm the presence of permafrost. In this last point it is important to highlight that the ground ice stored seasonally in active layer can have influence on the temperature of the springs. Therefore, it is highly recommended to carry out other methods. EC and TDS measurement and hydrochemical and isotopic analysis could provide valuable information.

We agree with the reviewer about the importance of integrating different methods to verify the role of permafrost on the hydrology of springs in mountain areas. However, based on our data collected in a sub-catchment of Val di Sole, only spring-water temperature proved to be effective in discriminating between permafrost affected and non-permafrost affected springs, whereas EC and isotopes where not useful (Carturan et al., 2016).

We agree with the reviewer that warm temperatures and cold temperatures alone are not sufficient to confirm respectively the absence and presence of permafrost. Warm temperatures of springs far from the main permafrost body (e.g. the Bordolona rock glacier) are already discussed in Section 5.3 (L. 533-538). Regarding cold temperatures due to possible seasonal ice in the ground, please see the reply to the following comment 2.6.

2.3) ERT profiles made in this work contribute to determining the presence or not of ground ice in the two rock glaciers studied. However, in one of them (Preghena) the resistive anomaly that would indicate the presence of ground ice is located at the 2D profile edges. In these sectors the reliability of the results decreases considerably. In addition, in this same rock glacier, a 20% error is considerably high.

We agree with the comment of the reviewer that the bottom and the edges of the tomogram are the least sensitive zones. We acquired a large number of measurements with the dipole-dipole multi-skip scheme (about 5200 quadrupoles) and, even after the applied filtering, the pseudo-section was homogeneously covered by apparent resistivities, including the edges and the bottom. We are confident we are not assessing artifacts, because the high resistive area is highlighted by both the ERT lines in the overlapping position (x<70 m in Line 1 and x>100 m in Line2). The data error of 20% applied in the inversion process was defined using the reciprocal analysis, which minimise possible inversion artifacts (Binley, 2015), and not the more common stacking error. Obviously, the expected data error can be estimated also with the stacking error (we acquired the measurements with a stacking range between 3-6, and a standard deviation threshold of 5%), nevertheless this approach usually overestimates the quality of the dataset and is less reliable (Binley, 2015).

We added these considerations in Section 5.4 (L. 577-579).

2.4) In general, the work is well structured, the reading is pleasant, the data is relevant, and the figures are of good quality and enriching.

Specific comments are made below.

Line 13: This work is not focused on the water contribution of the talus slopes. Nor are there many background to mention the importance or not of the talus slopes in the water supply to the rivers. Where does this statement arise?

Agreed, this can be confusing for the reader. We do not mention the talus slopes anymore and have slightly rephrased this sentence (L. 13-14)

2.5) Line 68-71: InSAR or DInSAR would not be useful techniques for this case since the relict or pseudo relict rock glaciers have no movement or the movement can be very slow and be in the same range as the uncertainty of the method. For this reason, since it is not a method applicable to this study, this sentence should be removed.

We have added these considerations in the Introduction, where we write about InSAR (L. 74-75). We would prefer keeping the reference to InSAR because as some readers may wonder if we have considered this methodology.

2.6) Line 73-75: The measurement could be at the end of summer or principles of autumn. The temperature trend in every summer should be considered, because with colder summers the thermal wave takes longer to enter the subsoil and completely defrost the active layer. This research methodology assumes that the water in which the temperature is being measured is influenced by ground ice. But, so that the temperature value registered in the springs is a real evidence of permafrost existence, it should not exist seasonal ice in the active layer.

The reviewer is right that cold temperatures can be due to the presence of seasonal ground ice. For this reason, we have stuck with the standard procedure of performing measurements at the end of the summer/beginning of autumn in order to exclude seasonal ground-ice influence on temperature measurements, or at least minimise its effect within the limit of our approach. It is true that seasonal ground ice formation could have a stronger influence on spring temperature after cool/short summer seasons, but this was not the case in the years we have analysed, characterised by warm/long summers. We added these considerations in the Discussion section (paragraph 5.3, L. 543-547).

2.7) Line 94-96: It should be specified that lithology is composed of each rock glacier. Considering that ERT has been applied, which are of the relict or pseudo-relict type, and that the amount of ice present can be small, the changes or gradients in the values of the electrical resistivity could be influenced by lithology. This information could also be added in section 3.4.

Agreed. The inspected rock glaciers are characterised by uniform lithology. Information regarding lithology was added in the text (Section 3.4, L. 244-245).

2.8) Line 156 (Data collection): Has a monitoring of temperature variation in springs during the day been performed? If there is a variation of temperature during the day this will influence the results depending on the measurement time.

Hourly spring water records collected during summer and early autumn using dataloggers show negligible variation of spring-water temperature during the day (Seppi, 2006). We added these considerations in Section 5.3 (L. 548-551).

Seppi, R.: I rock glaciers delle Alpi Centrali come indicatori ambientali (Gruppo Adamello-Presanella e settore orientale del Gruppo Ortles-Cevedale) - Rock glaciers of the Central Alps as environmental indicators (Adamello-Presanella Group and eastern sector of the Ortles-Cevedale Group). Phd Thesis, 199 pp., doi: 10.13140/RG.2.1.1186.5682, 2006.

2.9) Line 161-162: It is not completely clear to me. In the previous paragraph it is mentioned that 133 springs located downslope of rock glaciers were measured. This paragraph says that 67 (17 + 50) springs in rock glaciers were measured. What is the amount of spings in rock glaciers that have been measured?

Here we report how many rock glaciers we sampled, not how many springs downslope of rock glaciers. Before, we mention 133 springs downslope of rock glaciers because there were rock glaciers with multiple springs measured. We agree that this can be confusing and we have clarified it at the beginning of Section 3.2 (L. 165-166). Afterwards, at the beginning of Section 3.3 (L187-190), the manuscript already clarifies that we retained only one spring for each rock glacier sampled.

2.10) Line 175: How were those streamflows measured? With what criteria was that threshold (0.11 l/s) established?

We have a long experience of runoff measurements using the salt dilution method on mountain creeks and torrents. Based on this experience we were able to estimate by eye the runoff of springs, and to assess that a 0.1 l/s threshold was adequate for discriminating between semi-stagnant and well-fed springs, and thus to discard springs affected by large temperature fluctuations during the day. We have added that we estimated runoff with the aim of discarding springs with too low runoff at the end of Section 3.2 (L. 180-182). Please see also the reply to comment 1.6.

2.11) Line 251-252: In these cases of high contact resistance it is advisable to add under the sponge an aluminum foil of approximately 20 cm side and cover it if possible with fine material. If contact resistances remain high, after incorporating abundant salt water and aluminum foil, it is advisable to move the profile position to a more favorable sector. Personally, I do not recommend measuring with such high contact resistances.

We agree with the Reviewer that these were unfavourable conditions, even applying the traditional approach of stainless-steel spike coupled with sponges soaked in salt water (Hauck and Kneisel, 2008), which usually guarantee optimal contact resistances (<100 kΩ). In the specific case of the Preghena rock glacier, the surface was particularly dry due to drought conditions. In our experience, good quality ERT datasets can be collected on Alpine rock glaciers even with contact resistances larger than 100 kΩ, adopting large quadrupoles number acquisitions that considerably helps redundancy and data reliability, as we did. The surface of the rock glacier is almost entirely covered by blocks. For this reason, it was not possible to find a more favourable positioning.

2.12) Figure 8: The scale of values should be expressed in Ωm or kΩm, as in the text. In addition, it is convenient to add the RMS in each profile.

Agreed. We harmonised the units of measurement between the text and the figure. We have also added the RMS in each profile, as suggested.

2.13) Line 464 (section 5.3): One of the main uncertainties is that it is not known if there is ground ice from the previous winter that has been stored in an active layer and that is influencing the temperature of the springs. In a way, the monitoring of the springs over the years would allow to eliminate this situation to a certain degree.

Please see the reply to the previous comment 2.6. We added these relevant considerations in Section 5.3 (L. 543-547).

2.14) Another uncertainty factor is that water inside a rock glacier can follow different paths (Villarroel et al., 2022). In pseudo-relict rock glaciers, with the presence of ground ice in the form of islands, the water could follow paths without contact with the ice. I consider that this situations must be discussed.

We agree with the Reviewer that this situation must be included in the discussion. We added these considerations in Section 5.3 (L. 535), where we write about warm spring-water temperature due to the contact with unfrozen sediments: "or if permafrost is patchy and not in contact with groundwater paths".

---

## Author Response (AR2)

Dear Editor,

Thanks for your feedback on the latest version of our manuscript.

We definitely agree that it is important to carry out other methods in conjunction with the spring-water temperatures. For this reason, we have stressed in the paper that it is a preliminary/pilot/auxiliary method, not conclusive, which requires other investigations for confirmation.

We report this concept at lines 30-31 in the Abstract, 78-79 in the Introduction, 242 in the Methods, 509, 553 and 557-558 in the Discussion, 637 and 643-645 in the Concluding remarks (line numbers are referred to the latest version of the manuscript (4), track-change version).

Limitations and uncertainties in the spring-water temperature approach are deeply analysed in Section 5.3, with several integrations in response to reviewers' comments. In particular, we discuss the possible warm and cold bias that can affect spring-water measurements, and consequently rock glacier activity classification. We have added further considerations regarding the warm/cold bias and rock glacier classification at the end of Section 5.3 (line 558-560) and additional recommendation for further investigations (and possible examples) in the Concluding remarks at line 640-642.

As suggested, we have added discussions on the importance of the ice storage inside the rock glaciers of our study area, comparing our findings with those reported in the literature (lines 609-619). Additional references have been reported in the References Section.

Finally, we added a data availability statement (line 674-676) reporting the DOI of the open-access repository where the dataset used in this paper has been stored.

Thank you for consideration and kind regards,

Luca Carturan and co-authors.